# Digit-tracking as a new tactile interface for visual perception analysis

Guillaume Lio [1,2,3], Roberta Fadda [4,5], Giuseppe Doneddu[5], Jean-René Duhamel[1,2,6] & Angela Sirigu[1,2,6]*

Eye-tracking is a valuable tool in cognitive science for measuring how visual processing resources are allocated during scene exploration. However, eye-tracking technology is largely confined to laboratory-based settings, making it difficult to apply to large-scale studies. Here, we introduce a biologically-inspired solution that involves presenting, on a touch-sensitive interface, a Gaussian-blurred image that is locally unblurred by sliding a finger over the display. Thus, the user's finger movements provide a proxy for their eye movements and attention. We validated the method by showing strong correlations between attention maps obtained using finger-tracking vs. conventional optical eye-tracking. Using neural networks trained to predict empirically-derived attention maps, we established that identical high-level features hierarchically drive explorations with either method. Finally, the diagnostic value of digit-tracking was tested in autistic and brain-damaged patients. Rapid yet robust measures afforded by this method open the way to large scale applications in research and clinical settings.

---

[1] Institute of Cognitive Science Marc Jeannerod, CNRS, Bron, France. [2] University of Lyon, Lyon, France. [3] Reference Center for Rare Diseases with Psychiatric Phenotype Génopsy, le Vinatier Hospital, Bron, France. [4] Azienda Ospedaliera Brotzu, Cagliari, Italy. [5] Department of Pedagogy, Psychology, Philosophy, University of Cagliari, Cagliari, Italy. [6] These authors contributed equally: Jean-René Duhamel, Angela Sirigu. *email: sirigu@isc.cnrs.fr

In humans, vision is the dominant sensory modality to acquire knowledge about the external world. The structure of our retina is such that visual information can only be obtained by constantly moving our eyes in order to inspect visual details with the photoreceptor-dense central fovea. As a result, eye movement analysis provides a unique window into oculomotor, perceptual and cognitive mechanisms. Different methods have been developed to measure eye movements in humans and animals[1] such as electrooculography (EOG)[2,3], scleral search coil[4,5] and optical tracking[6,7] methods. Each of these techniques presents some advantages and limitations in regard to spatial and temporal resolution, invasiveness, complexity of operation, restrictions on freedom of movement (notably of the head). All have in common the ability to measure eye position directly and can be used to assess the function, integrity and neural control of the different classes of voluntary and reflexive eye movements.

In many applications of eye-tracking, investigators do not need detailed information about e.g., saccade dynamics, torsion or vergence components of the eye movements but are principally interested in knowing what an observer is looking at and in establishing the spatial and temporal allocation of processing resources over the visual scene[8–12]. Observed regularities in visual exploration patterns have helped identify universal aspects of perception like the preference for high-contrast boundaries, the avoidance of uniform or regularly textured surfaces[13], for specific items such as faces[14–16], and the decisive influence of internal goals and task instructions in prioritizing visual features[8,9,17,18].

Existing eye-tracking techniques confine such studies to laboratory tests conducted by trained technical or scientific personnel on a limited number of subjects. However, a fine characterization of statistical deviations from the norm across individuals or images, for e.g., diagnosis or design optimization in ergonomics and communication, requires large-scale measurements that are difficult to achieve with laboratory-based eye-tracking studies. In order to render this possible, we designed and tested a novel, mobile and connected solution for visual exploration analysis dubbed as "digit-tracking". The basic concept is to present a Gaussian-blurred image on a touch-sensitive display mimicking the low spatial resolution of the peripheral retina. By touching the display with a finger, the user unblurs an area equivalent to that of the foveal field just above the contact point location. The displacements of the fovea are thus inferred indirectly from the user's finger movements over the image. In order to validate this approach, we first compared visual exploration of inanimate, natural and social scenes by human participants whose gaze was monitored with digit-tracking and with the current gold-standard in eye-tracking, which consists in recording pupil displacements by means of an infrared sensor. Normalized probability density estimates of spatial exploration were used to compute attention maps and compare the two methods. We also trained a convolutional neural network (CNN) to predict the empirically derived attention maps and to compare how high-level visual features are hierarchically ordered during eye-tracking or digit-tracking image's explorations. The sensitivity of the eye and digit-tracking approaches for the diagnosis of autistic spectrum disorders (ASD) was assessed using attention maps obtained for 22 High-Functioning autistic patients who had undergone extensive rehabilitation and 22 matched control subjects. We reasoned that if a digit tracking-based procedure could detect individuals who had been trained and may have developed coping strategies to overcome their social interaction difficulties, it would make a strong argument for its potential as a diagnostic tool. Compared to the neurotypical population ASD patients' image exploration was characterized by avoidance of socially relevant features, and notably of gaze cues. Their attention maps deviated significantly from that of the control group, whichever method

considered, suggesting that digit-tracking represents a simple, rapid and practical alternative to direct eye-tracking for many large-scale applications. Finally, in further exploratory tests of the potential and sensitivity of digit tracking, we characterized the spatial exploration bias in 5 patients showing left neglect after right parietal cortex damage and we established that the method is suitable for visual attention studies in children from the age of 3 years, as well as non-human primates.

## Results

**Digit-tracking as a tool to measure attention-maps.** In order to measure visual exploration without an eye-tracking device, we presented on a touch-sensitive display a set of images deliberately degraded with an intensity close to the natural loss of acuity present in peripheral vision (Fig. 1a, b). Using this strategy, the image appears blurred to the subjects since the perceived acuity in their central vision is lower than the potential acuity of their foveal vision. However, the quantity of information discernible in peripheral vision is mostly unaffected by the procedure. When subjects touch the digital interface with their finger, an area just above the contact point is displayed with the full picture definition (Fig. 1a). This leads subjects to engage a digital exploration of the image in order to inspect in full resolution the regions where visual details need to be revealed. Subjects in fact spontaneously adopt an optimized exploration behavior which consists in an alternation between fast moves with slowing and direction changes around selected regions of interest. This exploration strategy is similar to the natural oculomotor behavior exhibited by humans when observing their environment, who constantly move their eyes in order to bring important areas onto the photoreceptor-dense central region of the retina. Using direct eye movement measurements, the identification of critical regions that are capturing visual attention is traditionally done using visual-attention maps calculated as the probability density of visual fixations on the images. Similar attention maps can be obtained from indirect measurements by calculating the probability density of the location of the center of the full-resolution aperture. Both approaches result in extremely similar visual-attention maps (e.g., Fig. 1c) with the same local maxima situated on the most perceptually salient areas of the observed image.

**Correlation between digit-tracking and eye-tracking.** Measuring eye position by tracking the movements of a finger over a blurred image could be susceptible to different bottom-up and top-down influences. For instance, the relative intensity of degradation of the simulated peripheral vision and the relative size of the simulated foveal area could have a strong impact on the exploration behavior. Also, digital exploration is slower and more energy-demanding than simple ocular exploration and it is possible that subjects exert greater cognitive control over their behavior with the finger pointing to where the eyes must look than with the eyes alone. In order to determine if, despite these potential issues, digit-tracking is a robust and reliable approach for studying visual exploration, a group of 22 subjects performed a simple picture exploration task under either digit-tracking or simple eye-tracking conditions. Each subject explored 122 pictures, divided into two sets (A and B) of 61 pictures. One subgroup of 11 subjects explored set A in the blurred-imaged, digit-tracking condition and set B in full-resolution condition with their eye movements being recorded using an infrared eye-tracking device. The second subgroup of subjects explored the same two sets of images but the assignment of recording method was reversed. Image sets, order of presentation and recording method were fully counterbalanced. For each image/subject couple, the probability density of exploration was revealed using kernel density estimates using fixation data weighted by fixations

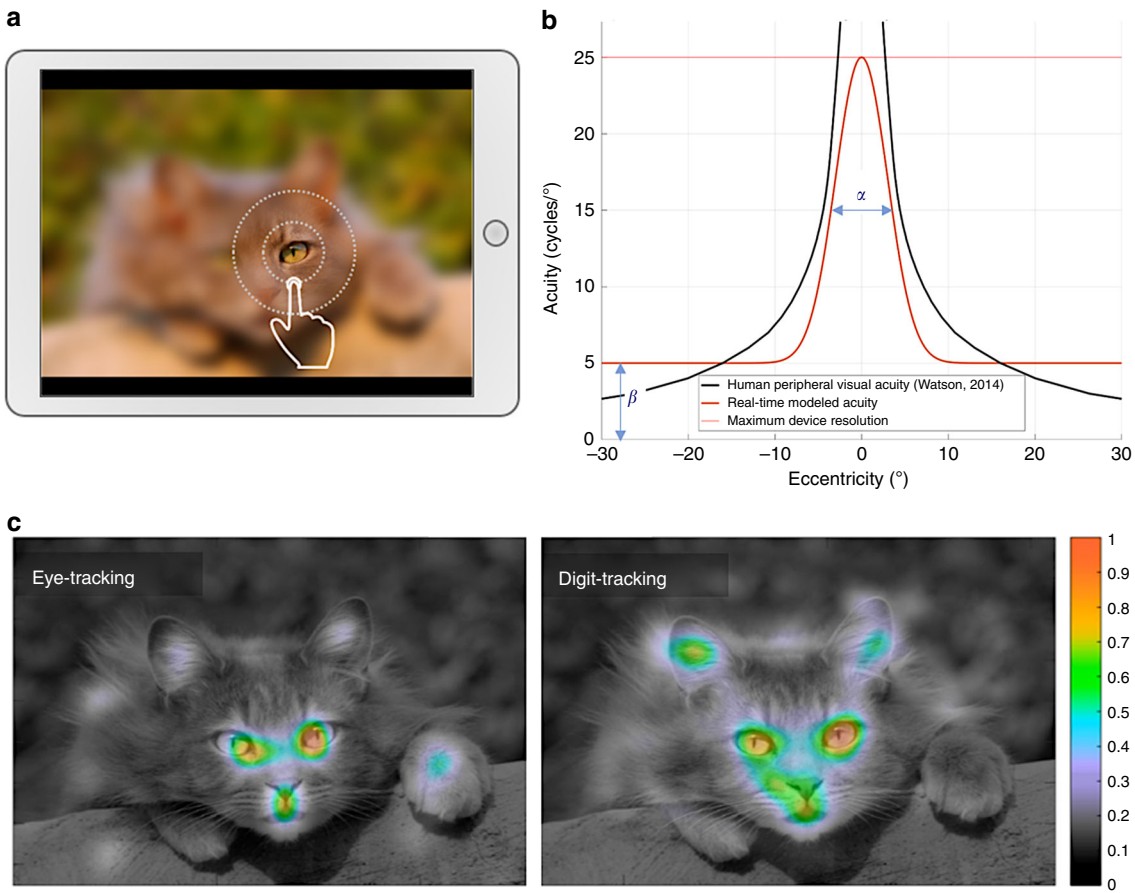

**Fig. 1** Digit-tracking method for recording ocular exploration via a connected touch-sensitive interface. **a** The resolution of the displayed picture is reduced using Gaussian blur, simulating peripheral vision, while in an area above the tactile contact point the central vision is simulated by revealing the picture in native resolution through a Gaussian aperture window. Subjects can explore the picture by moving their digit over the display. **b** Example of parameterization of the apparatus for real-time recording (orange curve). Two parameters are set: the size of the aperture window (α) and the maximum level of detail perceptible in the simulated peripheral vision (β). Optimal exploration is obtained using a simulated peripheral acuity close to the real human peripheral acuity (black curve–from ref. [55]) and a central vision slightly narrower than the foveal region (see Methods section for a complete description of the strategy for optimal adjustment of these parameters). **c** Average heat maps representing the picture's normalized probability density of exploration (scaled between 0 and 1), estimated by two independent groups of subjects (each group $N = 11$) using either eye-tracking (left) or digit-tracking (right) methods. The two estimates are highly correlated ($r_{\text{pearson}} = 0.73$) and show similar high attentional saliency regions.

duration for eye-tracking measurements, and the regularly sampled center of the simulated central vision for digit-tracking measurements. Our results show that both methods produce comparable attention maps that are significantly correlated (Fig. 2a; Set A: median correlation = 0.71, sign-test $p < 9 \times 10^{-19}$; Set B: median correlation = 0.72, $p < 9 \times 10^{-19}$; no significant differences between image sets: Wilcoxon $p = 0.72$).

Another informative measure of the noise sensitivity of the two approaches is the inter-subject correlation (ISC). ISC is calculated for each image and eye-tracking methods as the average Pearson correlation coefficient between the exploration densities of one subject with the exploration density of another subject. The higher the ISC, the more similar is the exploration across subjects and the better the technology to capture distinctive and idiosyncratic behaviors of a population. ISCs calculated with eye-tracking or digit-tracking are correlated (Fig. 2b; $r = 0.42$, $p < 2 \times 10^{-6}$), suggesting that ISCs are more picture than technology dependent. In other words, when a given image induces a particular exploration pattern, this pattern can be recorded with both methods. Furthermore, the absence of a significant difference for ISC between visual explorations measured with eye-tracking and digit-tracking (Fig. 2c; $\text{median}_{\text{eyes}} = 0.44$, $\text{median}_{\text{digit}} = 0.43$, Wilcoxon $p = 0.32$) indicates that image exploration is not more

sensitive to noise with the digital interface than with standard eye-tracking. A supplementary analysis considering the length of exploration shows that the observed correlations between the two methods show an initial rise and rapidly reach a stable point within the tested time window (Fig. S1). As a final check on the robustness of our method, we estimated the minimum number of subjects necessary to obtain stable and reproducible measurements. This estimate is obtained by calculating first, for each attention map recorded on a population of N subjects, the percent of variance that has been already explained with a population of N-1 subjects. Then, for each image and technology, the minimal number of subjects necessary to explain more than 95% of the total variance is determined (i.e., adding a supplementary subject will modify less than 5% of the variance of the estimated attention-map) (Fig. 2d). Whichever method is considered the same number of subjects is required to obtain reliable estimates of normalized exploration density (Fig. 2e; median number of subjects to explain 95% of variance = 5) and seven subjects are sufficient to explain more than 95% variance for most pictures used in this experiment. Using another "stability" metric proposed by Judd et al.[19] which measures how $n$ subjects predict the performance of $n$ other subjects, we confirmed the close parallelism between eye-tracking and digit-tracking performance as a function of number of

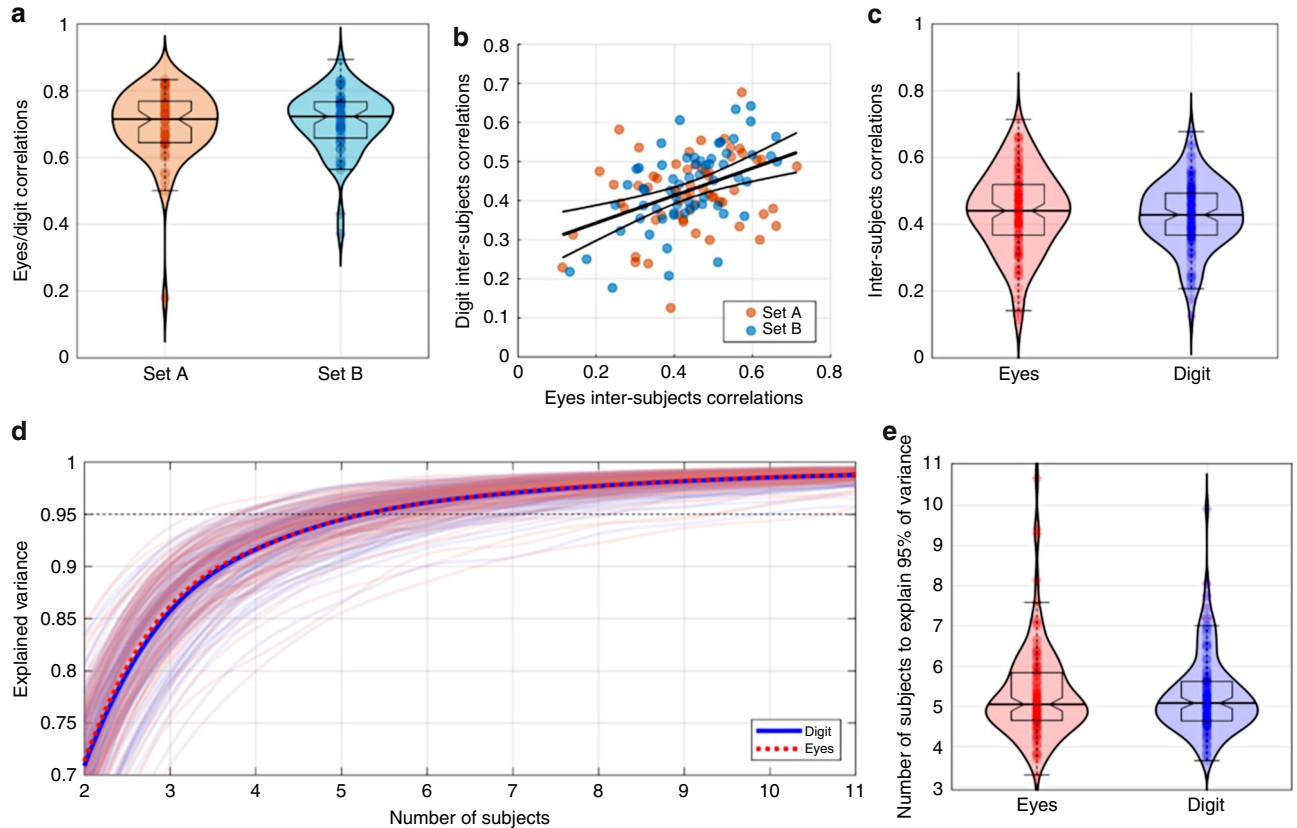

**Fig. 2** Digit and eye-tracking comparison. A total of 122 pictures featuring humans, animals, objects or abstract art were divided in two sets (Set A and Set B). Half of the subjects explored Set A using digit-tracking and Set B with eye-tracking, while the other half explored Set B using digit-tracking and Set A using eye-tracking, thus yielding two independent sets of attention maps for each method. **a** Violin plots of the Pearson's correlation coefficients calculated, for each picture, between the probability density estimates of exploration measured with eye-tracking or digit-tracking (11 subjects/condition). Both techniques can measure precise attention maps that are highly correlated (Set A: median correlation = 0.71, sign-test $p < 9 \times 10^{-19}$; Set B: median correlation = 0.72, $p < 9 \times 10^{-19}$; no significant differences between image sets: Wilcoxon $p = 0.72$). **b, c** Inter-subject correlations scatter and violin plots, respectively. Inter-subject correlations were calculated, for each image/technology couple, as the average Pearson correlation coefficient between the exploration densities of one subject with the exploration density of another subject. ISC calculated with eye-tracking or digit-tracking are correlated (B) ($r = 0.42$, $p < 2 \times 10^{-6}$). No significant differences can be found for ISC between explorations measured with eye-tracking or digit-tracking (**c**) (median$_{eyes}$ = 0.44, median$_{digit}$ = 0.43, Wilcoxon $p = 0.32$). **d** Convergence of exploration normalized density estimates. Each curve represents for one image/technology couple recorded with N subjects, the percent of variance of the exploration normalized density estimates that could be explained with N-1 subject. **e** Violin plots of the number of subjects necessary to explain more than 95% of variance. Eye-tracking and digit-tracking show similar performances (median = 5, seven subjects are sufficient to obtain stable measurements on most images).

subjects (Fig. S2). Together, these results suggest that our digit-tracking implementation has a reliability equivalent to that of traditional eye-tracking for the assessment of visual exploration.

Jiang et al.[20] used a similar biomimetic idea of a blurred image simulating the low spatial resolution of the peripheral retina and local deblurring, but with the computer mouse instead of the finger as user interface. The main objective of their study was to build saliency models that approximate an average viewer's performance, using the crowdsourcing marketplace to acquire a large number of image explorations. In order to compare this approach with digit-tracking and with eye-tracking, we acquired attention maps from a new group of 11 subjects using images from set A and the mouse exploration procedure described by Jiang et al. Mouse tracking was found to correlate well with eye-tracking, but performed significantly less well than eye-tracking and digit-tracking on measures of susceptibility to noise in individual subject data (Fig. S3).

**Deep-learning predicts feature hierarchy in attention maps.** Correlation analyses have shown that attention maps recorded

with eye-tracker and digit-tracking are substantially correlated, indicating that both reveal the same perceptually-salient areas in an image. However, the influence of the recording device on the levels of measured saliencies remains an important question since different recording environments could have different top-down influences on visual exploration. In other words, if we consider two perceptually salient features A and B, if feature A is more salient than feature B during a visual exploration recorded with eye-tracking, is this hierarchical order of saliency preserved with digit-tracking? As an example, we can consider the well-known bias toward the eye region[21]. Numerous eye-tracking studies have indeed demonstrated that the presentation of a static face is sufficient to trigger spontaneous non-verbal communications *via* the initiation of eye-contact behavior, even with a non-communicating agent[14–16,22]. This raises the question whether visual exploration with digit-tracking replicates this finding even though it is potentially more prone to conscious cognitive control.

An elegant solution to determine the level of saliency of the elements that make up a visual scene, is through the use of deep learning technology. Deep convolutional neural networks (CNN), thanks to their layer configuration that mimics the successive

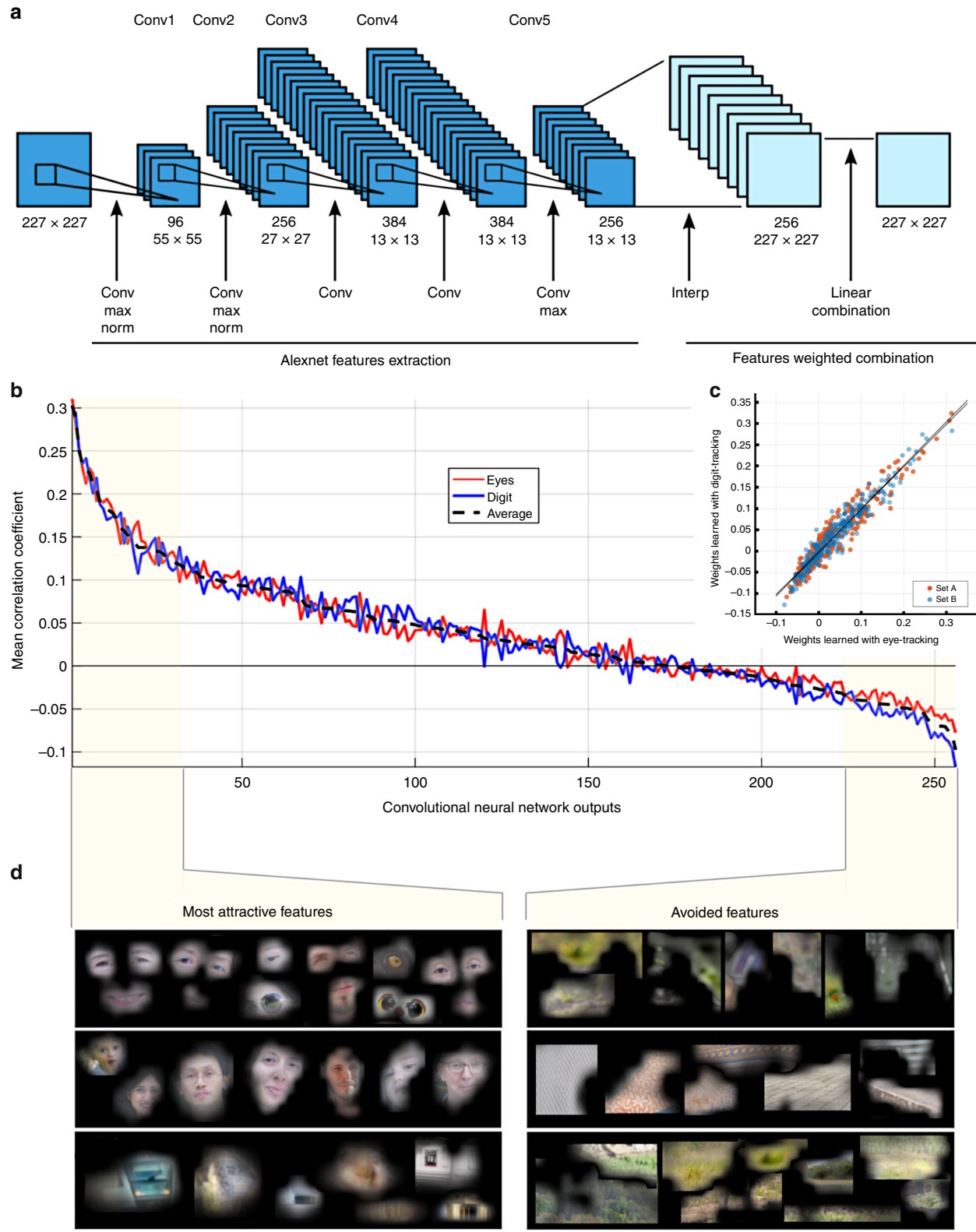

retinotopic processing stages that are carried out throughout the visual cortex, have the ability to decompose a single image in a set of numerous feature maps. Each feature map can be considered as a saliency map sensitive to a single low-level or high-level feature, depending on the depth of the considered layer. Then, after a learning phase with a set of recorded attention maps, these feature maps can be linearly combined to build a model of artificial vision

that can produce, from any image, a saliency map that predicts the visual exploration of a single subject or a population of subjects (Fig. 3a; Fig. S4). Crucially, inspection of the learned weights of the linear combination of the feature maps can reveal how subjects' visual exploration is optimized, by quantifying the contribution of each feature to the measured behavior. In other words, by relying on a very limited set of assumptions, it is

**Fig. 3** Convolutional Neural network computation of saliency maps. A model of artificial vision was used to predict the most salient areas in an image and test whether attention maps derived from digit-tracking and eye-tracking exploration data are sensitive to the same features in visual scenes. **a** Convolutional Neural Network architecture. The first five convolutional layers of the AlexNet[23] Network were used for features map extraction, and features are linearly combined in the last layer to produce saliency maps (e.g., Fig. S4). **b** Hierarchical ordering of learned weights in the last layer of the convolutional neural network (CNN). $X$-axis denotes the 256 outputs while the $Y$-axis denotes the mean Pearson correlation between an individual channel and the measured saliency map. Each channel can be seen as a saliency map sensitive to a single feature class in the picture. A strong positive correlation coefficient indicates a highly attractive feature while a strong negative correlation indicates a highly avoided feature. **c** Correlation between the weights learned using eye and digit-tracking (Set A $r_{pearson} = 0.95$, $p < 1 \times 10^{-128}$–Set B $r_{pearson} = 0.96$, $p < 1 \times 10^{-147}$). **d** High-level features are visualized by identifying in the picture database the most responsive pixels for the considered CNN channel. Example of the most attractive and the most avoided features corresponding to, respectively, the 3 most positively correlated and the 3 most negatively correlated channels of the CNN. Human explorations are particularly sensitive to eyes or eye-like areas, faces and highly contrasted details, while uniform areas with natural colors, textures, repetitive symbols are generally avoided.

possible to detect among the essential features that combine to create a large range of images (and our perception of these images), the hierarchical ordering of features that drive visual attention.

We used a well-validated convolutional neural network architecture for feature extraction (AlexNet Network[23]–Fig. 3a). The selected model was first trained to classify a set of more than one million images extracted from the ImageNet database (http://www.image-net.org) for the ImageNet Large-Scale Visual Recognition Challenge (ILSVRC[24]). Then, a second training phase was conducted in order to find the optimal linear combination of high-level feature maps that produce the saliency maps closest to observed human behavior (Fig. S4). Empirically-derived attention maps used for this second learning phase were computed from eye-tracking data in a first experiment and from digit-tracking data in a second one. The CNN architecture predicted human visual exploration behavior measured with either method (Cross-Correlation Score (CC) CCeye = 0.55; $p < 1 \times 10^{-24}$ sign test, CCdigit$_t$ = 0.63; $p < 1 \times 10^{-20}$ sign test–FWER corrected), and without significant differences in the quality of prediction ($p > 0.05$ paired $t$-test, $p = 0.037$ uncorrected paired sign-test (CCdigit > CCeye)). Strikingly, the hierarchical orderings of the 256 learned weights based on digit-tracking and eye-tracking measurements showed extremely strong correlations (Fig. 3b, Set A $r_{pearson} = 0.95$, $p < 1 \times 10^{-128}$-Set B $r_{pearson} = 0.96$, $p < 1 \times 10^{-147}$) (Fig. 3c). This indicates that whichever approach is used, the measurements are equally informative about the relative saliency of the elements in the visual scene (Fig. 3b). Concretely, elements like eyes, eye-like areas, faces and highly contrasted details, were found to be the strongest attractors of visual exploration whether measured by eye-tracking or digit-tracking. Conversely, elements that are generally avoided like uniform areas with natural colors, textures, repetitive patterns, etc. are at the bottom of the feature hierarchy (Fig. 3d). These results again confirm that eye-tracking and digit-tracking are equally reliable tools to explore attention biases in human visual exploration.

**Detection of atypical exploration behaviors**. The digit-tracking method can be implemented on any mobile and connected device, and being both user-friendly and calibration-free, it can offer clear practical advantages when tests must be conducted on particular subject populations or outside the laboratory environment. We therefore sought to explore its potential benefits for the detection and monitoring of visual exploration strategies in a clinical environment, where reliable quantitative evaluations of behavior are essential for the diagnosis and follow-up of diverse medical conditions. Impairments of visuospatial attention are a common consequence of vascular or traumatic brain injury. We tested patients with right parietal lobe damage and a suspicion of hemispatial neglect. Bedside examination with the digit-tracking system easily revealed the extent of the patients' left neglect (see the single case example in Fig. 4a and data for 4 other patients in Fig. S5).

Atypical visual behaviors are also observed in autistic spectrum disorders (ASD)[25–28]. This is illustrated in Fig. 4b showing the image exploration performance of a 14-year-old patient presenting clinical signs of ASD, with social avoidance and highly stereotyped, restricted, and repetitive patterns of behavior. As compared to an age-matched neurotypical subject, exploration of socially-rich images using the digit-tracking interface revealed a systematic tendency to avoid human figures, and a focus on non-social features. In order to further evaluate the relative sensitivity of digit-tracking and eye-tracking methods, we investigated visual exploration in 22 rehabilitated high-functioning autistic young adults using the same counterbalanced picture exploration procedure as above. The image sets contained both social and non-social images, allowing to assess and quantify atypical behavior without a priori assumptions about the diagnostic value of the different types of image. Stereotypical exploration patterns were obtained with both methods for pictures containing faces, revealing a strong attraction to the eye region in the neurotypical population and an exploration pattern avoiding this same region in autistic patients (Fig. 5a, see Figs. S6, S7 for further illustrations of exploration patterns). Further confirmation that this is a key distinctive marker of neurotypical and ASD visual behavior comes from the CNN framework (Fig. S8). Whereas for the neurotypical population, the top-ranking feature channel in the salience hierarchy corresponds to the eyes and face internal details ($p < 0.004$ for both recording methods, permutation test), these visual elements are significantly less attractive for ASD patients ($p_{eye} < 0.001$, $p_{digit} < 0.0001$, non-parametric rank test, FWER-corrected). Instead, the most powerful attractor of visual attention in ASD patient is a feature channel corresponding to the whole face area ($p < 0.004$, permutation test). These results indicate the tested ASD patients, like neurotypical subjects, pay attention to faces but underexplore finer facial cues. Importantly, the results obtained with the CNN analysis were highly consistent between eye-tracking and digit-tracking methods.

One last point to consider is whether this new tool is sufficiently robust and sensitive to have a practical utility in the diagnosis and monitoring of autistic spectrum disorders. In other words, how good is it at discriminating neurotypical from potentially pathological visual explorations and how does it compare with standard eye tracking? To this end, we calculated a simple 'exploration-neurotypicality score' based on the Z-transformed mean Pearson correlation between the attention map measured on a given subject and a set of normative attention maps measured on a control population (for more details about this calculation, see Methods section). This basic index detects anomalies in the explorations of ASD subjects with either recording method (respectively $p_{eye} < 0.0001$ and $p_{digit} < 0.00001$–Wilcoxon ranksum test) and yields correlated evaluations of neurotypicality (Spearman rho = 0.56 $p < 0.0002$, Fig. 5b). In order to compare the discriminatory power of the eye-tracking approach with that of digit-tracking, we first used a non-parametric distribution fitting tool to estimate the distributions of the score in both populations.

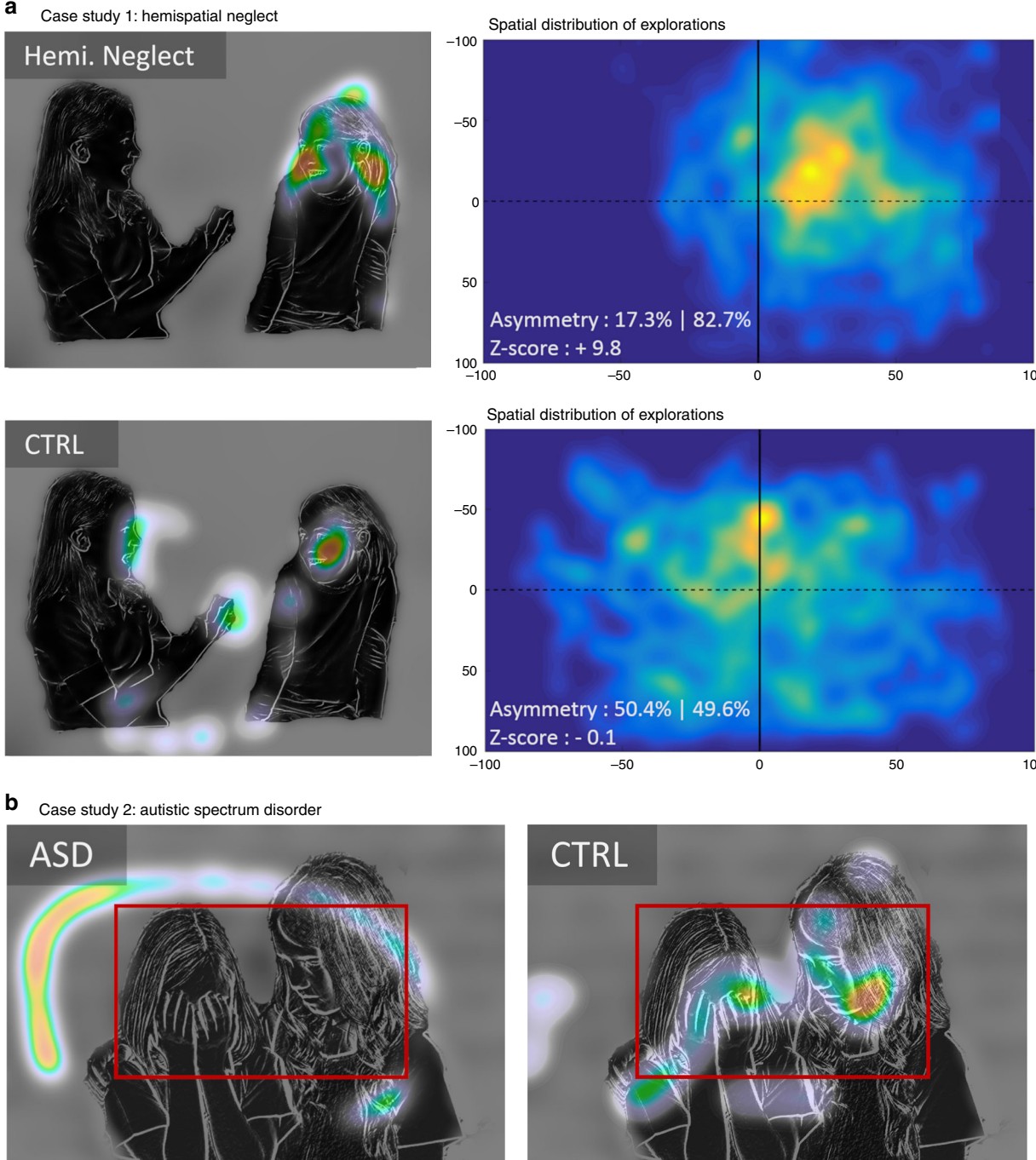

**Fig. 4** Atypical exploration behaviors recorded using digit-tracking. **a** Example of image exploration by a patient suffering from hemispatial neglect (Hemi. Neglect) due to right-parietal lobe damage following stroke and a control subject. The right panels represent the average spatial distribution of the explorations recorded for 32 images in a single examination session. The spatial attention bias can be precisely quantified: for the control subject, 49.6% of their exploration is on the right side of the display (corresponding to a Z-score of −0.1, centile 45, according to a reference population, $N = 22$), whereas for the patient, 82.7% of the exploration has been recorded on the right side of the display (corresponding to a Z-score of 9.8, centile > 99.999). **b** Exploration of an image with social content by a 14-year-old non-verbal autistic child compared to a neurotypical control subject. The patient adopts an exploration strategy which avoids human faces (red frame), whereas these are the most explored scene elements in the control population. Please note that original faces have been modified to hide individuals' identity.

Then, we calculated the Receiver Operating Curves (ROC) derived from the estimated distributions to finally obtain evaluations of the classification performances using the AUC (Area Under the Curve). AUCs were estimated at 86.2 and 90.8% with Eye-tracker and with Digit-tracking, respectively (Fig. 5c). These high performances indicate that even a simple score derived from non-optimized blurred images explored with a finger on a tablet can be used to identify ASD patients through their visual behavior with a sensitivity and a specificity equivalent to that which can be obtained with classical eye-tracking.

**Digit-tracking in children and non-human primates**. The present validation of digit-tracking in the assessment of visual

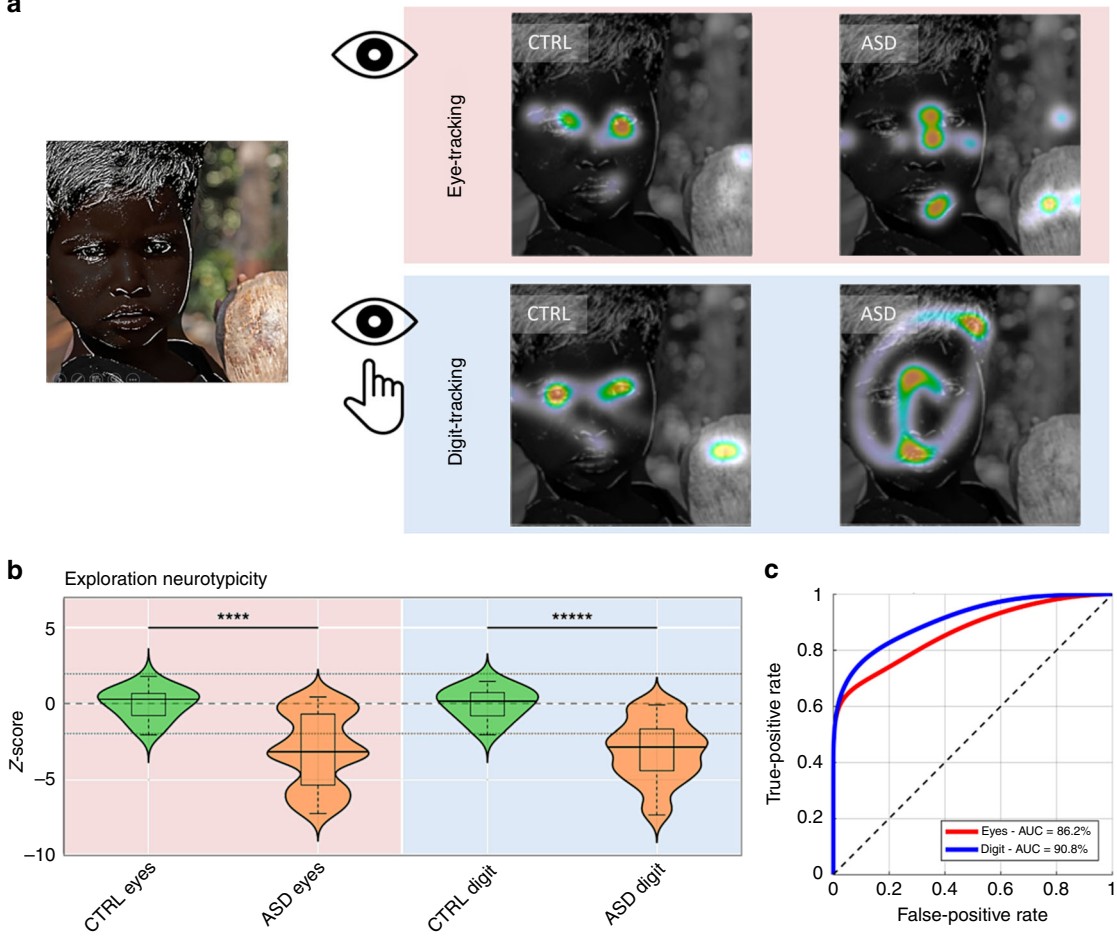

**Fig. 5** Detection of abnormal visual explorations in autistic spectrum disorders (ASD). **a** Typical exploration behaviors observed in four subjects (two ASD and two control subjects), recorded with either eye-tracking or digit-tracking. High-functioning autistic patients (ASD) tend to adopt an atypical face exploration strategy that avoids the eye area. Note similarities in the explorations, even if they are recorded with different methods on different subjects. **b** Results of the group analyses: For each subject, a single score was calculated to quantify the neurotypicity of attention maps obtained in patients ($N = 22$) and control subjects ($N = 22$). Both methods (Eye-tracking and digit-tracking) detect anomalies in the attention maps of the ASD population (respectively $p_{eye} < 0.0001$ and $p_{digit} < 0.00001$–Wilcoxon rank-sum test) and are correlated (Spearman rho = 0.56 $p < 0.0002$). **c** Receiver Operating Characteristic (ROC) curves for ASD/CTRL classifications with the 'exploration-neurotypicality score'. Please note that original faces have been modified to hide individuals' identity.

attention potentially opens broad perspectives for research in visual cognition. It is therefore of interest to establish the suitability of digit-tracking for populations other than adult human subjects. We have initiated a developmental study assessing visual performances in children from several age groups. Preliminary results indicate that children as young as 3 years-old can explore images using the digit-track interface and show well-defined attention maps (Fig. S9). We also conducted tests in a macaque monkey who had access within its home environment to a touch sensitive display running the digit-track software. In order to encourage spontaneous image exploration, the monkey was simply rewarded when a pre-determined track-path length had been reached, with no time constraint. The monkey learned to interact with the device and showed expected attraction to significant features in social scenes (Fig. S10).

## Discussion

Our results show that a rather simple biomimetic approach, implemented on a common computing device equipped with a touch-sensitive display, enables measurements of human visual exploration behavior with satisfactory performances. We show that the measurements obtained with digit-tracking are highly reliable

and comparable to those obtained with optical eye-tracking. These measurements are not biased by the method of acquisition, which requires using a motor effector not normally used during natural visual exploration. Despite the necessarily different dynamics of the two types of exploration, the hand having much greater inertia than the eye, the attention maps computed from digit-tracking and eye-tracking data are strongly correlated, at both the individual and the group levels. In a sense, this is not so surprising since the eyes and fingers often move in a synergistic way, assisting one another as in children learning to read or when performing manual actions that require high precision. As an indirect measuring device, digit-tracking is obviously excluded from some applications of optical eye-tracking, like calculating the dynamics (e.g., speed, acceleration) of reflexive and voluntary eye movements (e.g., optokinetic and vestibulo-ocular reflexes, smooth pursuit, pro-saccade and anti-saccade, oculomotor capture, etc.), blink rate or pupil size. But even within the confines of visual perception and attention analysis, for which it was designed, one could rightly ask what are the benefits of this method, given that increasingly sophisticated and portable methods exist to directly measure ocular movements with high accuracy.

The principal value of digit-tracking lies in its extreme simplicity and portability. A digit-tracking experiment can be

implemented on any recent device equipped with a touch-screen interface, such as a tablet or smartphone. The system is always ready to record data, needing no calibration and it can be used in almost any environment without requiring a data-acquisition specialist or complex, cumbersome, and potentially anxiety-provoking equipment for the subject. Implemented on a cloud computing architecture, data acquisition can be highly parallelized and distributed on many research structures, clinical centers and different populations. Although less portable than a connected mobile device, one can also use a standard computer display and mouse interface to compute attention maps from blurred image explorations. We compared the Jiang et al. mouse-tracking implementation with eye-tracking and digit-tracking and found that mouse-tracking achieves, on average, reasonably good correlations with eye-tracking but is more susceptible to noise in the individual data as indicated by lower inter-subject correlation values than digit-tracking. Although this would require further confirmation, one reason possibly explaining the better performance of digit-tracking is that it takes advantage of the fact that, compared to mouse manipulation, eye-hand coordination is a developmentally early, natural, highly practiced and accurate motor synergy[29]. The difference in performance between the two implementations also speak to the difference in the goals pursued in the two study. The Jiang et al. method was aimed mainly at identifying salient image features in large image data bases and did so by averaging out measurement noise across many subjects. In contrast, our goal is to achieve single-subject measurements that are as noise-free and reliable as possible (i.e., equivalent or better than eye-tracking) so that it could be used not only in group studies but also for individual diagnostic purposes.

As a novel method of measurement of the human cognition, digit-tracking can be very beneficial for at least three research domains: cognitive, computer and medical sciences. In each of these areas, the digit-tracking approach allows to expand the horizon of research and bring a level of flexibility almost impossible to reach in the necessarily highly-controlled laboratory environment. Indeed, for ethical and practical considerations, most experiments that involve eye movement recordings are limited in scope, duration, number of subjects and number of repetitions. This can lead to problems in generalizing results to other populations, stimuli and experimental contexts[30,31]. With a connected and distributed recording environment, large trans-cultural studies can be easily and rapidly conducted and replicated at a small cost, measurements can be made on a broad range of visual stimuli, effect sizes can be evaluated rapidly before launching large-scale studies or more controlled eye-tracker experiments, etc.—paving the way to more robust and generalizable results.

Computer science and cognitive sciences are mutually beneficial when they work together. Many advances in image and signal processing come from the understanding of neurophysiological principles and their transposition by humans into machines designed to solve engineering problems. Examples of biomimetic approaches in the domain of artificial perception are numerous, from the Young-Helmholtz theory of trichromatic color vision[32–34] that spurred the invention of the color camera to the most recent developments in deep learning with convolutional neural network that are mimicking the layers of the visual cortex[35], via the Perceptron[36] and the solving of multi-sensors mixing problems through Blind Source Separation[37]. The recent innovations have pushed computing capabilities to a level at which the latest biomimetic algorithms need to optimize a large number of parameters and resort to huge learning databases. Consequently, acquiring human behaviors on a new scale becomes mandatory. Classification algorithms need the labeling of a great number of images and acquiring such data is easily

feasible through a WEB interface. But when some regression problems need to be solved, such as saliency map prediction, more complex behaviors need to be recorded. Saliency models can be used to overcome the limited spatial resolution of convolutional neural networks, they can be trained to solve specific object detection tasks and help neuroscientists understand how humans can effortlessly detect targets in their visual field using highly degraded peripheral vision. However, due to a limited quantity of data available, even the most recently developed saliency models are mainly sensitive to bottom-up information and fail to model top-down modulations and to detect semantic information that guides selective attention[38,39]. Large-scale data acquisition using digit-tracking could contribute to the development of computational models of visual attention that achieve or even surpass human performances. Such models could in turn serve to investigate cognitive impairments involving specific alterations or bias of attention, like hemispatial neglect, autism or other neurodevelopmental, psychiatric and neurodegenerative disorders.

In this study, we showed how digit-tracking coupled to a computational approach can be used to analyze visual exploration in neurotypical and ASD subjects. Modeling the obtained data with a convolutional neural network revealed how subjects allocate their attentional resources as they explore pictures with social content, what key high-level visual features drive the attention in the two populations. Notably, we identified a deficit in the exploration of regions containing eyes, which is partially counterbalanced by a preserved exploration of other face regions. This result replicates the atypical face exploration behavior often reported in the ASD population[25–28]. It should be emphasized that the patient population tested consisted of highly rehabilitated young adults whose visual exploration behavior might reflect successful learning of social adaptation strategies. As the development of innovative strategies to improve social interactions in patients with ASD focus on eye–contact or eye-contact-like behavior[40,41], digit-tracking could prove useful in a clinical setting to detect, quantify and follow the evolution over time of this behavior.

We have designed digit-tracking from the outset to take oculometry out of the laboratory and into medical environments where, in most industrialized countries, biological testing and neuroimaging are commonly available but reliable tools to accurately measure and quantify behavior are lacking. Standard neuropsychological testing is invaluable in the assessment of cognitive performance in many pathological conditions, from developmental disorders to dementia. But such tests suffer from test-retest reliability issues, can be easily biased, anxiety provoking and usually require good language comprehension. By contrast, spontaneous visual exploration is a natural behavior that can be elicited in patients of all ages without any instruction and effort and be of high diagnostic value. Indeed, eye movements during normal visual exploration are often stereotyped and little contaminated by noise. But optimal exploration depends on the integrity of cognitive abilities and alterations in cognitive function, such as those present in e.g., neglect or autism, can be easily detected. Thanks to its portability and ease of use, the digit-tracking technique could be used to extend and conduct fine grained evaluation of visual exploration to other disabilities for diagnosis, follow-up and rehabilitation. This new approach also opens further perspectives in developmental research, where measures of spontaneous visual orientation are commonly used to assess early perceptual and cognitive competences (see e.g., refs. [42,43]). Finally, the digit-tracking approach is perfectly suited for non-human primate research and could be useful in the development of novel non-invasive techniques for behavioral training and cognitive assessment.

To sum up, we believe that this simple, calibration-free and connected solution for human visual cognition research is ideally suited for large-scale longitudinal studies, evaluations of therapeutic strategies, differential diagnosis, early detection, and would certainly facilitate testing in environments where technical and scientific and medical expertize is not easily accessible. These features also make digit tracking a potentially highly useful tool in the evaluation of visual material in applied science domains like design, ergonomics, user experience and communication.

## Methods

**Participants**. Two groups of participants were recruited for this study. One group of 22 subjects with no history of psychiatric or neurological disease (referred to as neurotypical or control (CTRL) group) served in the comparative evaluation of digit-tracking and eye-tracking technologies and as sex-matched and age-matched controls for ASD patients. The ASD group was composed of 22 male patients (mean age: 20, σ: 2.5) recruited at the Azienda Ospedaleria Brotzu (Cagliari, Italy) and with a clinical diagnosis of autism according to the *Diagnostic and Statistical Manual of Mental Disorders, 5th edition*[44]. They were all rehabilitated patients with a large range of intelligence estimates (WAIS: range [58–141], mean: 96, σ: 22 (IVth edition−2008)) and autistic symptoms intensity (ADOS[45]–Mod IV: range [2–14], mean: 8.2, σ: 3.4) at the time of the study. All participants had normal or corrected to normal vision.

**Ethics committees approval**. Tests conducted in human participants were approved by French (Sud-Ouest, project N° 2018-A02037–48) and Italian Ethical Committees (Azienda Ospedaliero-Universitaria of Cagliari, project N°AOB/2013/1, EudraCT code 2013–003067–59) and prior to the inclusion in the study, a written informed consent was obtained from all participants and/or their legal representative, as appropriate. Tests conducted on non-human primates were in conformity with current guidelines and regulations on the care and use of laboratory animals (European Community Council Directive No. 86–609) and were conducted under a research protocol authorized by the French Ministry of Research Ethics board (APAFIS n° 2015061213048343).

**Picture database**. Two sets of 61 pictures each (Set A and Set B, N = 122) representing natural and social scenes, humans, animals, objects and/or examples of abstract art were selected. The image database was deliberately varied in order to not preclude any exploration behavior by the tested populations. Each picture contained one or several salient features and was optimized for a 1280 × 1024 pixels screen resolution. The type and global content of the scenes were matched between the two subsets. The picture data base contains images depicting human bodies and faces because these stimuli induces characteristic exploration patterns that provide the best basis for methods comparisons and because these exploration patterns are also strongly diagnostic of autistic subjects' gaze behavior.

**Experimental procedure**. The experiment consisted in recording picture explorations from the two populations using two recording devices: a standard infrared video-based eye-tracker and digit-tracking (both are described below). Each participant was instructed to freely explore each picture during a single session lasting approximately 30 min, one picture set using direct eye-tracking (≈15 min), and a second picture set with digit-tracking (≈15 min). Order of recording method and picture set (A or B) assignment were fully counterbalanced between subjects within each group.

**Direct eye-tracking method**. Eye position was recorded by means of an infrared eye-tracker (Tobii 1750, sampling rate = 50-Hz), following a 5-points calibration routine. Recordings were made by experienced specialists familiar with the equipment used and with the running of cognitive experiments in normal and patient populations. The stability of the calibration and the behavior of the subject was constantly checked by the operator during the experiment by, in real time, observing if the fixations on the fixation-cross was correctly realized and observed before each picture presentation. If any problems were detected during the recording, the data were excluded from the analysis in order to not bias the quality and stability of measurements. Using these procedure and experimental setup, the usually observed mean deviation between real and measured focalization point is in the order of one degree of visual angle[46]. The Clearview 2.7.0 software was used for picture presentation, eye-gaze data recording and pre-processing. The experiment took place in a quiet, dedicated room. Pictures were displayed on 33.8 × 27.4 cm video monitor (1280 × 1024pixels) placed at a distance of 50 cm from the subject's eyes (0.87°/cm–39° × 31°). Each trial began with the presentation of a fixation cross, prompting the subject to press a key in order to display a picture. The fixation cross stayed on the screen for 1 s and was replaced by a 4 s picture presentation. Due to a technical problem, data from two ASD patients during exploration of the set B images were not included in the analysis of eye-tracking data.

**Digit-tracking method**. Digit-tracking was implemented on a Microsoft Surface Pro 4 computer, configured as a tablet with a 2736 × 1824 pixels touch interface (267 Pixels Per Inch) and a physical dimension of 26 cm × 17.3 cm (46.5° × 31° of visual angle—0.56°/cm—considering a median eyes-screen distance of 32 cm, range 27 to 37 cm). A Matlab (r2016a–the MathWorks, Inc.) custom code, designed using the Psychophysics Toolbox Version 3[47–49], was used for picture presentation, real-time picture modification and data recording. The subject was sitting at a table and could manipulate the device without any constraint. Each trial began with the presentation of a virtual button at the center of the screen which the subject had to touch in order to launch picture presentation. The picture appeared blurry on the device so as to simulate the low acuity of peripheral vision. When the subject touched the screen, an area above the contact point was displayed with the picture's native resolution. Removing the finger from the display restored the image's fully blurred appearance and sliding the finger across the display caused the full resolution area to be updated so as to track in real time the finger's current position. Thus, the subject could simultaneously engage in a process of ocular exploration guided by the finger. Since exploring with the finger and eye is slower that with the eyes only, the time of exploration is not limited. Instead, the cumulative distance of exploration was calculated and exploration ended when a predetermined threshold was reached.

In our implementation of the digit-tracking technology[50], four parameters are set: the level of picture degradation needed to simulate peripheral acuity, the size of the simulated foveal area, the location of the foveal window with respect to the contact point and the length of exploration. Peripheral vision was simulated using a Gaussian blur filter of size σ = 40 pixels (≈0.7°) and the foveal vision was simulated using a Gaussian aperture windows of size σ = 110 pixels (≈1.9°), centered on the contact point's coordinates but shifted upward by 80 pixels (≈1.4°) to prevent masking by the subject's fingertip. In order to determine exploration length, we conducted a preliminary eye-tracking experiment using the same physical display surface as in this study (2736 × 1824 pixels) and with 71 different pictures presented for 6 s to 8 subjects. The recorded mean path-length was 850pixels/second of exploration (25th centile = 725pix/s; 50th centile = 821pix/s; 75th centile = 969pix/s). Considering that the duration of presentation of stimuli for the current eye-tracking experiment was set to 4 s, we choose to stop digit-tracking exploration when a track-path of length 4000 pixels (≈68° of visual angle) had been reached (1000pix/s–17°/s), in order to ensure an equivalent quantity of exploration in the two experiments.

**Comparisons between digit-tracking and eye-tracking**. Several analyses were conducted to compare visual exploration measures obtained with digit-tracking and eye-tracking methods. First fixations were identified in the continuous eye-gaze data using a fixation filter (Fixation radius = 50 pixels, minimum duration = 50 ms). Only fixations in the picture area were considered for analysis. Then, for digit-tracking, the regularly sampled location of the center of the simulated foveal windows was transposed in the original picture space to ensure that all analyses are done in the same space with the same spatial resolution for the two methods (picture maximized to fit a 1280 × 1024 pixels rectangle). For each subject and each picture, the probability density of exploration, or attention map, was calculated from the fixations data using kernel density estimation with a Gaussian kernel weighted by fixation duration and a bandwidth of 30 pixels (≈1° of visual angle) in order to compensate for the usual imprecisions encountered with these methods (e.g., ref. [46]). Finally, each density was scaled between 0 and 1 by dividing the whole density by the maximum density value and group estimates were calculated by averaging scaled densities recorded at the subject level and scaled using the same unity-based normalization procedure.

The comparison between attention maps measured with eye-tracking and with digit-tracking was performed using a correlation analysis. For each picture and each recording method, a group attention map was computed by averaging the attention map across subjects, yielding two group-attention maps by picture, derived from two independent populations with two different recording method. Then, for each picture, the similarity between eye-tracking and digit-tracking measurements was evaluated using Pearson correlation for a total of 61 correlation coefficients for each dataset, and tested using non-parametric statistical hypothesis tests. Thus, for each dataset, the median correlation was tested using Wilcoxon rank-sum test (Bonferroni-corrected for the 2 comparisons) and the difference between the two median correlations was tested with uncorrected Wilcoxon rank-sum test.

The sensitivity to the noise of a recording method can be estimated using a counterbalanced inter-subject correlation analysis. Inter-subject correlations were calculated, for each image/technology couple, as the average Pearson correlation coefficient between the attention maps of one subject with the attention maps of another subject. The higher the inter-subject correlation (ISC) the more similar is the exploration between subjects. The linear relationship between ISC calculated with eye-tracking and with digit-tracking was evaluated with Pearson correlation, and the difference was tested using Wilcoxon rank-sum test.

A key consideration in the building of an experiment is to estimate the minimal number of subjects necessary to obtain a stable and reliable measurement of the population behavior. In order to investigate this specific point, a convergence analysis was designed to estimate the minimal number of subjects necessary for each recording method, to obtain stable attention maps. For each picture, and each

recording technology, a convergence curve was obtained using the following algorithm:

- Choose a random permutation of N subjects.
- Compute N attention-maps, estimated by averaging from 1 to N subjects.
- For $n = 2$ to N, compute the percent of variance explained by n-1 subjects to the attention map computed with n subjects.
- Repeat the steps 1, 2, and 3, 40 times, and average the 40 curves obtained.

This procedure yields one convergence curve per picture and these curves can then be averaged to obtain a convergence estimate for each technology (Fig. 2d). Then, for each picture and each technology, the minimal number of subjects necessary to explain 95% of the variance were determined and compared across technology using non-parametrical statistical testing and distributions analyses (Fig. 2e).

**Convolutional neural network processing of attention maps**. Analyzing the similarity between attention-maps obtained with two different measurement devices with a similarity score like the Pearson correlation is a simple and a necessary primary approach that is commonly used to evaluate the quality of a measurement or a prediction of an image exploration behavior. But, correlation scores can be sensitive to the method used to calculate attention maps and to the type of pictures used during the experiment. In order to evaluate more precisely the similarities and differences between classical eye-tracking and digit-tracking regarding how subjects allocate their attentional resources on the images, we develop a second method to compare attention-maps based on a convolutional Neural Network Architecture. This architecture was used to automatically analyze the image content and evaluate among the identified elements those that attract more or less the human eye. To this end, the Matlab (Mathwork Inc.) implementation of the AlexNet[23] Network was selected. The images and recorded attention maps were cropped and scaled to match the input resolution of the network ($227 \times 227$ pixels). The AlexNet network was then modified to learn how to solve the following regression problem: predicting measured attention maps from input images—in other words, to make a saliency prediction. Weights learned to solve this regression problem are quantitative indicators of how each image element contributes to how subjects distribute their attentional resources on the images.

To predict saliency maps measured with eye or digit-tracking, the pre-trained AlexNet[23] Network was used to generate maps sensitive to various high-level visual features. The selected model was trained on a subset of more than one million images, extracted from the ImageNet database (http://www.image-net.org) for the ImageNet Large-Scale Visual Recognition Challenge (ILSVRC)[24]. Dedicated originally for large-scale image classification, the CNN architecture was transformed using the Matlab Neural-Network toolbox (Mathwork Inc.) to solve a regression problem and generate saliency predictions. First, the outputs of the fifth convolutional layer, rectified by a linear unit (ReLU layer), were interpolated to obtain 256 feature-maps with the same resolution of the input image ($227 \times 227$ pixels). Then the maps were linearly combined using a weighting function to generate the predicted saliency. This approach can be considered as a simplified implementation of the Kümmerer et al. method for saliency map prediction[51]. The optimal weighting function was learned using a small database of 122 couples of image/saliency map and validated with LOO (Leave One Out) cross-validation. Two networks with the same architecture were trained, one using eye-tracking data, and the second using digit tracking data. The quality of the prediction was evaluated by Pearson correlation (CC score) using the data that was not included in the training phase. Then, CC scores obtained with eye-tracking (CCeye) and digit-tracking (CCdigit) were tested using non-parametric sign test (Family Wise Error Rate (FWER) corrected) and compared using paired $t$-test and uncorrected paired sign-test. Two learning methods were tested for computing the optimal weights: wCorr and SVMregress. wCorr is a fast estimate of the weights by averaging the Pearson Correlation Coefficients calculated between a feature-map and the measured saliency map (the 256 weights are learned independently). SVMregress is a more standard learning strategy using SVM (Support Vector Machine) regression to linearly combine feature maps (e.g, refs. [51–54].). A learning strategy optimized for high-dimensional data was selected considering the important number of parameters estimated and the correlation in the data after the interpolation step (256 regression parameters, learned using fitrlinear–Matlab r2017a (Mathwork Inc.)–with a 10-fold cross-validated lasso regularization). Both are equivalent in terms of prediction quality (Pearson Correlation = 0.63 for both methods for digit-tracking prediction–Fig. S4B). Nevertheless, the first method (wCorr) was selected due to a better consistency in the learned weights (Fig. S4C), suggesting a faster convergence and therefore more stable results when the learning image database is small.

In order to test whether the two recording methods produce equivalent measurements of saliency, we used the previously described CNN architecture to quantify how much each of the 256 high-level visual features hierarchically contributes to the explorations measured by eye-tracking or by digit-tracking. First, the 61 pictures of the first dataset (set A) were selected to train the network using the group-level attention maps measured on the first population of 11 neurotypical subjects with eye-tracking. Then, the same pictures were used to train the network again, but using the group-level attention maps measured on the second population of 11 neurotypical subjects evaluated with digit-tracking. The similarity between the learned weights was assessed using Pearson correlation (Fig. 3c). Lastly, this procedure was replicated using the 61 pictures of the set B.

In order to visualize the hierarchical ordering of the contribution of the 256 visual features, the weights were averaged across image datasets first, then across recording methods and ordered in decreasing order (Fig. 3b). The features with the highest positive weights are those that are the most attractive and conversely, the features with the most negatives weights are the most avoided ones. Thus, these 256 weights can be viewed as a signature of how subjects hierarchically consider high-level visual features to optimize their visual explorations. Finally, in order to visualize these features, pixels in our image database that are activating the most the corresponding features are selected and displayed to assess more intuitively which image elements are the main contributors of the measured visual exploration strategies.

CNN was also applied to visual exploration data from the ASD population. Two types of analysis were conducted on the exploration data recorded with eye-tracking and digit-tracking in the group of ASD patients. The CNN architecture was used to identify the high-level visual features that are the best predictor of the measured explorations in ASD patients and compare these to the same data obtained in the control (CTRL) population of neurotypical subjects. Also, for each subject and each recording method, an 'exploration-neurotypicality' score was computed in order to test whether digit-tracking is as efficient as eye-tracking at detecting atypical exploration behavior in the ASD population.

**Single subject performance prediction using CNN**. Since the adopted CNN architecture generates stable estimates of how a population prioritizes high-level visual features guiding their visual exploration, the same architecture was trained to predict the visual exploration behavior of each subject and for each recording method. To do so, 256 parameters were learned for each couple subject/recording method. These 256 parameters represent a signature of how each subject has hierarchically considered high-level visual features during the recorded visual explorations (Fig. S8A). Then, in order to identify the most salient features for a given population, four mean signatures of 256 parameters were estimated with four groups of subjects: ASD with Digit-Tracking, ASD with Eye-tracking, CTRL with Digit-Tracking and CTRL with Eye-Tracking. The first null hypothesis tested was that the most salient feature is different for different recording method. If the most salient features are identical for Eye-Tracking and Digit-Tracking, the null hypothesis is rejected with a p-value equal to 1/256 ($p < 0.004$). This hypothesis was tested on the ASD and on the CTRL population (two tests, significant if $p < 0.025$ after Bonferroni correction) (Fig. S8A). Post-hoc analyses were then conducted on saliency weights of identified high-level visual features. One analysis was conducted on the feature identified as most salient in the CTRL population (Fig. S8B; Channel 3/256–A channel sensitive to internal facial cues) and another one on the feature identified as most salient in the ASD population (Fig. S8C; Channel 229/256–A channel sensitive to faces). The tested null hypotheses were that the weights estimated for a feature are identical for the two populations. Due to the non-gaussianity of the tested parameters, non-parametric tests have been used (Kruskal-Wallis ANOVA and Wilcoxon rank-sum test—all results are corrected to control the Family Wise Error Rate (FWER)).

**Exploration of "neurotypicality"**. For each subject, a single score was calculated to quantify how much recorded attention maps are similar to, or deviate from those of the reference neurotypical population. For each explored picture, we first calculated a score obtained by averaging the correlation coefficients between a subject's attention-map and the attention-maps calculated for 10 independent neurotypical subjects. These values were then Z-transformed according to the neurotypical reference population, averaged across pictures, and Z-transformed again. This procedure thus yielded a simple 'exploration-neurotypicality score' for each subject and exploration recording device. Differences between these scores were assessed using non-parametric Wilcoxon rank-sum tests. The correlations between scores evaluated with eye-tracking and digit-tracking were evaluated using Spearman correlations. Then, to estimate the power of discrimination of the 'exploration-neurotypicality' score, the distribution of the latter measurement for each of the four studied populations (CTRL with eye-tracking, CTRL with digit-tracking, ASD with eye-tracking, ASD with digit-tracking) were estimated using a non-parametric distribution fitting tool (Gaussian Kernel density of $\sigma = 0.6$), the receiver operating characteristic (ROC) curves for ASD/CTRL classification were then calculated with the fitted distributions for both recording methods and finally the corresponding areas under the curves (AUCs) were evaluated to quantify the distance between the ASD and the CTRL derived distributions.

**Reporting summary**. Further information on research design is available in the Nature Research Reporting Summary linked to this article.

## Data availability

Data on healthy subjects that supports the findings of this study and Matlab scripts are available on Open Science Framework http://osf.io/x5ryp. Patients' data are available from the corresponding author on reasonable request.

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

## Acknowledgements

This work was supported by Centre National de la Recherche Scientifique, Labex Cortex University of Lyon I ("Investissement d'Avenir") Grant ANR-11-LABEX-0042 (AS, JRD), ANR-13-BSV4–0010–01 (JRD) and SATT Pulsalys CNRS (A.S., G.L., J.R.D.). We thank Stephane Donnet for continuous support and Victoire Mironneau and Pénélope Lacombe for help in testing subjects.

## Author contributions

G.L., J.-R.D. and A.S. designed research; G.D. and R.F. recruited patients and performed clinical evaluation. G.L. wrote software and analyzed the data. G.L., J.-R.D. and A.S. wrote the paper. All authors discussed data and approved the paper.
