## [Peer Review File · Nature Communications]

Reviewer #1 (Remarks to the Author):

In this manuscript the authors report on a novel methodology called digit tracking, which they developed to take 'visual attention research out of the laboratory'.

To do so, they ingeniously set up a system, which uses blurred images that can be unblurred by means of touch similar to how oculomotor behavior helps to temporarily unblur aspects of the external environment by moving them into foveal vision.

The authors convincingly demonstrate that the new methodology of digit tracking yields results that are comparable with established eye tracking techniques both by means of correlation analyses and by means of deep learning that focuses on the features, which drive learning.

In addition they report data from a study in adults with high-functioning autism. Here, digit tracking is shown to replicate well known findings of autism-related differences in visual exploration of faces and other social scenes.

Finally, the authors report individual cases of patients with brain damages, children and a monkey to lend additional anecdotal support to their method.

In my opinion, the paper is extremely well-written and the new methodology is very interesting in providing similar findings to eyetracking. The limitations of the new technique are discussed (saccades, etc.), but not at great length.

My main question is whether this methodological development really represents a major step forward for visual attention research. I appreciate that the method is portable, but recent advances in smartphone technology will likely make it possible to do eyetracking on mobile devices as well. In my view, the paper is very much a methodological paper that - at the content level - does not present any particularly new insight into autism or visual attention.

I would also be keen to see functional neuroimaging studies, which demonstrate commonalities and differences at the level of the brain between eye and digit tracking to give us a more precise idea of what exactly is similar/different across both methods.

Reviewer #2 (Remarks to the Author):

Review of "Digit-tracking": A tactile interface for visual perception analysis by Lio et al.

This paper presents a method for assessing the deployment of visual attention to images that does not require video-based eyetracking, by allowing participants to locally deblur an image using a touchscreen. The paper presents evidence that priority maps thus obtained are similar to eye gaze densities (recorded via eyetracking) in both neurotypical participants and autistic participants. The paper also presents some other case studies using digit tracking (a patient with hemispatial neglect, young children and a macaque monkey). The authors also present an analysis of the deep neural network features correlated with priority maps in both eyetracking and digit tracking, arguing that they are similar.

I think the proposed digit tracking technique will allow some interesting applications where traditional eyetracking is too costly, at least for measuring densities of visual attention maps. I believe the general conclusion of the paper, that this technique can be used to estimate spatial attention maps in lieu of eyetracking. However, as well as a number of methodological questions (see below), my three main problems with this paper are

1. Lack of appropriate integration with existing literature. The core contribution of the paper – the digit tracking technique – is extremely similar in motivation and implementation to that used in the SALICON dataset. SALICON allows images to be locally deblurred in a way that also mimics retinal acuity falloff, but uses a computer mouse rather than a touchscreen. While this is less portable (and thus the current implementation is more useful for e.g. clinics, infants and animals), it has the advantage that it can be deployed on the web more easily. The SALICON paper (Jiang et al, 2015) does just that, collecting eyetracking data from 16 lab subjects and a large online sample (using Amazon's Mechanical Turk) of 10,000 images annotated by 60 observers each. The authors do not cite this paper. While digit tracking is a nice complementary technique, the SALICON method does detract substantially from the novelty of the current approach.

An additional point regarding existing literature integration: The deep neural network model the authors employ is, as far as I can see, almost exactly the architecture used by Kummerer et al (Deep Gaze I). The authors cite this paper but don't point out that their CNN regression model is the same architecture.

2. I'm unsure what is really learned by the CNN analysis, beyond that the maps are similar (which has already been shown in Figure 1 and 2). I disagree with specifics of interpreting deep features. The paper adopts the view that the high layers of a deep neural network are selective for single high-level image features (e.g. Figure 3 caption "Each channel can be seen as a saliency map sensitive to a single feature in the picture"). However, this is clearly not the case for CNNs. High layers combine successive nonlinearities rendering easy interpretation intractable. Picking maximally-activating image patches does not necessarily mean a unit's activation is "caused" by whatever is in the patch – hence the large literature on interpretability metrics for CNN features (see e.g. <https://distill.pub/2017/feature-visualization/>, Kim et al (2017)).

3. Because of (1) and (2), I'm unsure what novel scientific insights the paper provides. The digit tracking technique seems like a useful proxy that could yield interesting new possibilities for data collection. This paper shows some proof of concepts of this. But if I try to summarise what I now know after reading this paper, I am left with "digit tracking on a touchscreen could be used to measure spatial priority maps as a complimentary technique to eye tracking and mouse tracking (SALICON)." The lack of new scientific insights make me feel this paper may have greater impact at a journal focussed on methods.

Specific comments / questions

* N = 11 seems like a very low number of subjects to be able to reliably estimate inter-subject correlation.

* The Convergence analysis does not show how many subjects are needed to obtain "stable attention maps" as written, but rather is another measure of inter-subject consistency. If two subjects performed exactly the same, this would jump up to 100% with only one subject. A better analysis to show how "stable" the attention maps are is to not normalise to the total variance, but rather show how the slope of improvement of a performance metric (e.g. correlation or AUC) changes as more subjects are added. If the metric isn't changing much then that would be evidence that the maps are stable; if the AUC is still increasing after adding the 11th subject, this is evidence that more subjects are required to obtain stable maps. Something like this was presented in Judd et al (2012; Table 1). My worry with the results presented is that readers may erroneously think that they can make reasonable conclusions after measuring data from only 5 subjects.

* The paper continuously uses the word "hierarchical" to describe the results of the CNN analysis. The CNN is hierarchical but the analysis presented uses only one layer; the authors rather seem to use this to refer to their rank-ordered linear weights for the different feature maps. This is just a linear combination, which is in no sense "hierarchical" (e.g. the second weight is not determined from the first weight).

- * p. 7 "it is possible to detect among the essential features that combine to create almost all possible images (and our perception of these images)". This is overstated. How does AlexNet (1) create almost all possible images and (2) explain how the "essential features" combine to create "our perception of these images"?
- * What does it mean if a high-functioning autistic person is "rehabilitated"?
- * Discussion: The discussion on page 10 and 11 veers into rambling incoherency. For example, the Young-Helmholtz trichromatic theory is cited as an "example in the domain of artificial perception"?
- * p.13: Eyetracking accuracy not discussed. I am worried from the example in Figure 1 that the calibration accuracy was not very good (e.g. Figure 1C, heatmap is not centered on the cat's eye)
- * p. 14: A "probability density" sums to 1, but the densities used in the study are normalised to have a max of 1. Correct language throughout paper.
- * Fixation densities were estimated with a kernel bandwidth of 30 pixels for both eye data and digit data. How was this selected? How do the results depend on this choice? Given that the screen resolution on the touchscreen was far higher than the eyetracking video monitor, does the 30 px bandwidth correspond to the same visual angle in both cases, or different? If different, does this create a problem for the comparison?
- * p. 15: "The sensitivity to the noise of a recording method can be estimated using inter-subjects correlation analysis" -- this analysis would confound the noise of the method (precision of estimating spatial attention location) with the consistency of the subjects. You could have a very precise measurement device but a low inter-subject correlation if people are inconsistent (or vice versa, consistent subjects measured with a poor device will show lower correlation than they should). Re-word.
- * The optimisation of the linear weighting function for combining AlexNet feature maps contains more parameters than data. Discuss.
- * p. 17: "This network was trained twice, using empirically derived eye-tracking data first, then using digit tracking". This sentence reads as if it's a single network being trained sequentially on both datasets, but I think rather you've trained two independent networks?
- * p. 19: "non-parameteric distribution fitting tool" -- how is it non-parametric if it assumes a Gaussian kernel density?
- * Figure 2C: in the inter-subject correlations, it seems that each subject contributes more than one data point -- which would make these distributions no longer independent. Discuss in relation to the statistical tests of correlation coefficients / rank-tests, which assume independence.
- * Figure 5: caption is confusing, and could be misunderstood to mean that the plots in B show data from only four subjects.
- * Figure 5C: if I understand correctly the classification is based on 22 ASD (20 for Eyes condition) and 22 Ctrl subjects. If this is the case, I would expect the ROC curves to be less smooth -- they should jump up / right as threshold changes.
- * Figure S1C: The analysis of the SVMregress coefficients seems misguided. The Lasso penalty forces all the weights to be really small (compare scales of top and bottom scatterplots). So all these weights are basically just at a point in the middle of the top plot -- ie. they've all been set to near-zero. In this case, I'm not even sure why this model works at all - it seems the regularisation penalty is too strong.

Open data and materials

The primary data, materials and code upon which this manuscript is based do not seem to be publicly available. I strongly encourage the authors to either

1. Make the data and materials publicly available in a reliable third-party repository (for example, the Open Science Framework -- see <http://osf.io/> or Zenodo (<http://zenodo.org/>)).

OR

2. State in the manuscript method section their reason(s) for not doing so.

If the authors choose to do (1), there are many more details and guidelines on how to do this at <https://opennessinitiative.org/the-initiative/>. Not only is this a great move in support of open science and transparency, but it may even mean the paper gets cited more (see for example Piwowar & Vision, 2013 or McKiernan et al 2016). It will certainly facilitate any future secondary analyses.

If the authors choose (2), note that there are many legitimate reasons why (1) cannot be done. For example, the authors may not be able to effectively anonymise data, or they do not own the copyright to the data or materials. In these cases, analysis and experiment code can almost always be released, but the degree to which the authors comply is entirely up to them. I require only that they state in the manuscript why they have not.

Should the authors be unwilling to do (1) or (2), then I cannot recommend the paper for publication because it does not meet the minimum quality requirements for a scientific manuscript.

References

Jiang, M., Huang, S., Duan, J., & Zhao, Q. (2015). SALICON: Saliency in Context. In 2015 IEEE Conference on Computer Vision and Pattern Recognition (CVPR) (pp. 1072–1080). Boston, MA, USA: IEEE. <https://doi.org/10.1109/CVPR.2015.7298710>

Judd, T., Durand, F., & Torralba, A. (2012). A benchmark of computational models of saliency to predict human fixations. CSAIL Technical Reports. Retrieved from <http://dspace.mit.edu/handle/1721.1/68590>

Kim, B., Wattenberg, M., Gilmer, J., Cai, C., Wexler, J., Viegas, F., & Sayres, R. (2017). Interpretability Beyond Feature Attribution: Quantitative Testing with Concept Activation Vectors (TCAV). ArXiv:1711.11279. <http://arxiv.org/abs/1711.11279>

McKiernan, E. C., Bourne, P. E., Brown, C. T., Buck, S., Kenall, A., Lin, J., ... Yarkoni, T. (2016). How open science helps researchers succeed. *ELife*, 5. <https://doi.org/10.7554/eLife.16800>

Piwowar, H. A., & Vision, T. J. (2013). Data reuse and the open data citation advantage. *PeerJ*, 1, e175. <https://doi.org/10.7717/peerj.175>

Reviewer #3 (Remarks to the Author):

The authors introduce a new tactile interface for visual perception analysis. This new method is clever and seems to have potential value in a restricted number of domains.

My most important concern reflects the lack of open science of this project. The whole manuscript reads like a brochure for a new software tool and, given the patent that was filed, it seems like a matter of time before this technology is transformed into a commercial product. Especially because this manuscript is pre-dominantly a methodological contribution (new theoretical is lacking), I would only want to advise Nature Communications to publish this type of paper if the software is open source and freely available. To me, open source is key in bringing the scientific endeavor forward.

Besides this main point, I think the authors should be more clear in stating for which cognitive domains this technique is interesting. It's definitely not a replacement for eye tracking as a whole (which is how the manuscript is currently framed), because eye tracking is widely used in all types

of domains. Here, scene viewing is convincingly demonstrated, but I don't see this technique play a role in measuring oculomotor competition, such as in the antisaccade or oculomotor capture task, or visual constancy, such as double-step and landmark tasks.

The authors report a reliable correlation between eye-tracking and digit-tracking, but only present the image for 4 seconds (in the eye-tracking version). Because of experimental constraints (e.g. central fixation point), exploration starts at the centre of the screen. It should therefore be checked whether this correlation also holds for the final two seconds and is not driven by the initial movements. They should also examine whether measures of search efficiency correlate between both tracking methods.

The case descriptions provide little information. If the authors really want to make a case for e.g. visuospatial neglect, they should test a full patient group.

Response to Reviewers

Reviewer #1 (Remarks to the Author):

In this manuscript the authors report on a novel methodology called digit tracking, which they developed to take 'visual attention research out of the laboratory'. To do so, they ingeniously set up a system, which uses blurred images that can be unblurred by means of touch similar to how oculomotor behavior helps to temporarily unblur aspects of the external environment by moving them into foveal vision. The authors convincingly demonstrate that the new methodology of digit tracking yields results that are comparable with established eye tracking techniques both by means of correlation analyses and by means of deep learning that focuses on the features, which drive learning.

In addition, they report data from a study in adults with high-functioning autism. Here, digit tracking is shown to replicate well known findings of autism-related differences in visual exploration of faces and other social scenes. Finally, the authors report individual cases of patients with brain damages, children and a monkey to lend additional anecdotal support to their method.

Q1: In my opinion, the paper is extremely well-written and the new methodology is very interesting in providing similar findings to eyetracking. The limitations of the new technique are discussed (saccades, etc.), but not at great length.

R1. We thank the reviewer for his enthusiastic assessment of the present study. We agree that our technique is an alternative to eye-tracking in the restricted context of visual scene exploration analysis. It is indeed important to establish explicitly the scope and limitations of digit-tracking and we expanded the relevant section of the Discussion:

“As an indirect measuring device, digit-tracking is obviously excluded from some applications of optical eye-tracking, like calculating the dynamics (e.g. speed, acceleration) of reflexive and voluntary eye movements (e.g. optokinetic and vestibulo-ocular reflexes, smooth pursuit, pro- and antisaccade, oculomotor capture, etc.), blink rate or pupil size.”

Q2: My main question is whether this methodological development really represents a major step forward for visual attention research. I appreciate that the method is portable, but recent advances in smartphone technology will likely make it possible to do eyetracking on mobile devices as well. In my view, the paper is very much a methodological paper that - at the content level - does not present any particularly new insight into autism or visual attention.

R2. It is indeed possible that one of our method's main advantages - its portability - will fade away when smartphone evolution integrates eye-tracking technology. For the moment, the most advanced system implemented in a smartphone is a function of the ARKit2 Dev Kit for the iPhone X, which, in fact, is not a true eye-tracking but 3D face-tracking with interpolation of the line of sight. It would allow the user to move a cursor on the screen, but its precision is low and needs an active adaptation of the user to compensate tracking errors and is therefore not well suited as a measuring instrument for research or clinical applications.

Q3: I would also be keen to see functional neuroimaging studies, which demonstrate commonalities and differences at the level of the brain between eye and digit tracking to give

us a more precise idea of what exactly is similar/different across both methods.

R3: This could be an interesting question for future studies but clearly beyond the scope of this paper which focuses on measurement performance and potential applications of digit-tracking.

Reviewer #2 (Remarks to the Author):

Review of “Digit-tracking”: A tactile interface for visual perception analysis by Lio et al.

This paper presents a method for assessing the deployment of visual attention to images that does not require video-based eyetracking, by allowing participants to locally deblur an image using a touchscreen. The paper presents evidence that priority maps thus obtained are similar to eye gaze densities (recorded via eyetracking) in both neurotypical participants and autistic participants. The paper also presents some other case studies using digit tracking (a patient with hemispatial neglect, young children and a macaque monkey). The authors also present an analysis of the deep neural network features correlated with priority maps in both eyetracking and digit tracking, arguing that they are similar.

I think the proposed digit tracking technique will allow some interesting applications where traditional eyetracking is too costly, at least for measuring densities of visual attention maps. I believe the general conclusion of the paper, that this technique can be used to estimate spatial attention maps in lieu of eyetracking. However, as well as a number of methodological questions (see below), my three main problems with this paper are

Q1: Lack of appropriate integration with existing literature. The core contribution of the paper – the digit tracking technique – is extremely similar in motivation and implementation to that used in the SALICON dataset. SALICON allows images to be locally deblurred in a way that also mimics retinal acuity falloff, but uses a computer mouse rather than a touchscreen. While this is less portable (and thus the current implementation is more useful for e.g. clinics, infants and animals), it has the advantage that it can be deployed on the web more easily. The SALICON paper (Jiang et al, 2015) does just that, collecting eyetracking data from 16 lab subjects and a large online sample (using Amazon's Mechanical Turk) of 10,000 images annotated by 60 observers each. The authors do not cite this paper. While digit tracking is a nice complementary technique, the SALICON method does detract substantially from the novelty of the current approach.

R1: We thank the Reviewer for pointing out this study of which we were unaware, which reflects our insufficient familiarity with the engineering literature on artificial vision. Both our and Jiang et al.’s methods (Jiang et al., 2015) are based on the biomimetic idea of using a blurred image to simulate the poor sensitivity of the peripheral retina and a dynamically updated full-resolution sub-region the size of the fovea. The mode of implementation of the image degradation/restoration is different between the two methods but the fundamental principles are quite similar.

During the early development of our method in 2014 we tested several solutions and finally settled on an optimized configuration based on a tactile, rather than a mouse interface. The mouse tracking approach yielded measurements with an accuracy lower than those obtained with eye tracking and this is also what Jiang et al. observed (see their Figure 6). As we understand it, the primary goal of the SALICON project was to use the crowdsourcing

marketplace in order to acquire a large number of image explorations. The mouse tracking approach they adopted to do so may be adequate for building of saliency models that approximate an average viewer's performance, but it is not designed nor intended to measure visual exploration behavior at the subject level and analyze individual differences. It should also be emphasized that, unlike our study, Jiang et al. did not conduct experiments comparing mouse and eye tracking directly, but instead compared mouse tracking maps with an existing eye tracking data set.

In our preliminary work, only the digit tracking method reached the precision of reference, i.e. an accuracy at least equivalent to eye-tracking acquisitions. There may be several reasons for the better performance of digit over mouse tracking, that pertain to the image degradation method, testing procedure and, of course, the use of a tactile interface, which takes advantage of the fact that eye-hand coordination is a developmentally early, highly practiced and accurate motor synergy (Esteve-Gibert and Prieto, 2014). However, we find that it is better to respond to the Reviewer's concerns with data rather than with mere arguments. Therefore, we have designed and tested an implementation of the SALICON mouse tracking technique and compared it to digit-tracking. We followed the method described in the Jiang et al. paper regarding mouse display, behavior and exploration time. We used our own image degradation algorithm in order to ensure that any observed difference between mouse- and digit-tracking would be imputable only to the mode of exploration. Eleven new subjects were tested on a subset of 61 images from our original study (Set A).

As expected from the Jiang et al. results, mouse-tracking yielded exploration maps that were correlated with eye-tracking maps, with an average eye-mouse correlation coefficient a little over $r=.70$, which is in the same range as the eye-digit average correlation reported in our study. However, two notable differences were observed:

(1) Mouse tracking induced a central bias, unrelated to image content, in the exploration maps. We believe this is an artifact imputable in part to the fact that, in the SALICON implementation, the location of the mouse pointer needs to be visible to the user and it is materialized by a red circle that always appears at the center of the computer monitor when a new image is displayed. Of course, this initial position effect can be eliminated by discarding the first mouse "fixation", which is what we did prior to performing any statistical analysis on these data. Some central bias nevertheless remained present, suggesting that there is a tendency for subjects to bring the mouse cursor back to center during exploration, a behavior that we do not observe with digit-tracking.

(2) More critically, the analyses that we performed to assess the susceptibility of each method to noise in individual subject data show that mouse-tracking performs significantly less well than the eye-tracking gold standard and also less well than digit-tracking. As can be seen in the attached figure, inter-subject correlations, a measure of how consistent exploration maps are across subjects, do not differ between eye- and digit-tracking (data from Set A of our original manuscript) but those obtained with mouse-tracking are significantly lower than either eye- or digit-tracking (Figure 1A). Furthermore, the estimated minimum number of subjects necessary to converge on stable and reproducible measurements (i.e. necessary to explain more than 95% of the total variance) is significantly higher for mouse-tracking than eye- and digit-tracking (Figure 1 B, C).

These differences between digit- and mouse-tracking implementations speak to the difference in the philosophy behind the two approaches. SALICON was aimed mainly at identifying

universally salient image features in large image data bases and does so relatively well by averaging out measurement noise across many subjects. In contrast, our goal was to achieve single-subject measurements that are as noise-free and reliable as possible (i.e. equivalent or better than eye-tracking) so that it could be used not only in group studies but also for diagnostic purposes. The two techniques are clearly related, but what needs to be emphasized is that our method and our parameter settings were designed to achieve the latter goal and this is what makes our implementation substantially different from SALICON.

In the revised manuscript, the SALICON anteriority is acknowledged at page 6:

“Jiang et al.¹⁹ used a similar biomimetic idea of a blurred image simulating the low spatial resolution of the peripheral retina and local deblurring, but with the computer mouse instead of the finger as user interface. The main objective of their study was to build saliency models that approximate an average viewer’s performance, using the crowdsourcing marketplace to acquire a large number of image explorations. In order to compare this approach with digit-tracking and with eye-tracking, we acquired attention maps from a new group of 11 subjects using images from set A and the mouse exploration procedure described by Jiang et al. Mouse tracking was found to correlate well with eye-tracking, but performed significantly less well than eye-tracking and digit-tracking on measures of susceptibility to noise in individual subject data (Fig. S3).”

The comparison between our implementation of SALICON and digit-tracking is presented in Supplementary Figure S3, and the results discussed in the Discussion section:

Supplementary Figure S3. Eye-, digit- and mouse-tracking comparison. A total of 61 pictures featuring humans, animals, objects or abstract art were used (Set A). Mouse-tracking was implemented using the same image degradation parameters as for digit-tracking but the method implementation was otherwise identical to Jiang et al.’s (2015). A) Plots of the Pearson’s correlation coefficients calculated, for each picture, between the probability density estimates of exploration measured with eye-tracking, digit-tracking or mouse-tracking (11 subjects/condition). The first two techniques can measure precise attention maps that are highly correlated

between subjects, but mouse-tracking shows significantly lower inter-subject correlations than eye-tracking ($p < 5.4 \times 10^{-4}$ sign test uncorrected) or digit-tracking ($p < 4.9 \times 10^{-5}$ sign test uncorrected). **B)** Convergence of exploration density estimates. Each curve represents for one image/technology couple recorded with N subjects, the percent of variance of the exploration density estimates that could be explained with $N-1$ subject. **C)** Plots of the number of subjects necessary to explain more than 95% of variance. Eye-tracking and digit-tracking show similar performances and require fewer subject than mouse-tracking to obtain stable measurements (eye v. mouse: $p < 1.9 \times 10^{-5}$ sign test uncorrected; digit v. mouse: $p < 6.5 \times 10^{-5}$ sign test uncorrected). These differences between the performance of the digit- and mouse-tracking implementations could reflect the objectives pursued by the two approaches. The mouse-tracking technique of Jiang et al (2015) was aimed mainly at identifying universally salient image features in large image data bases and does so relatively well by averaging out measurement noise across many subjects. In contrast, our digit-tracking was developed and optimized to achieve single-subject measurements that are as noise-free and reliable as possible (i.e. equivalent or better than eye-tracking) so that it could be used not only in group studies but also for individual diagnostic purposes.

Discussion section:

“Although less portable than a connected mobile device, one can also use a standard computer display and mouse interface to compute attention maps from blurred image explorations. We compared the Jiang et al. mouse-tracking implementation with eye- and digit-tracking and found that mouse-tracking achieves, on average, reasonably good correlations with eye-tracking but is more susceptible to noise in the individual data as indicated by lower inter-subject correlation values than digit-tracking. Although this would require further confirmation, one reason possibly explaining the better performance of digit-tracking is that it takes advantage of the fact that, compared to mouse manipulation, eye-hand coordination is a developmentally early, natural, highly practiced and accurate motor synergy (Esteve-Gibert and Prieto, 2014). The difference in performance between the two implementations also speak to the difference in the goals pursued in the two study. The Jiang et al. method was aimed mainly at identifying universally salient image features in large image data bases and did so by averaging out measurement noise across many subjects. In contrast, our goal is to achieve single-subject measurements that are as noise-free and reliable as possible (i.e. equivalent or better than eye-tracking) so that it could be used not only in group studies but also for individual diagnostic purposes.”

Q2: An additional point regarding existing literature integration: The deep neural network model the authors employ is, as far as I can see, almost exactly the architecture used by Kummerer et al (Deep Gaze I). The authors cite this paper but don't point out that their CNN regression model is the same architecture.

R2: We now acknowledge the similarity with Kummerer in the relevant paragraph of the Methods section:

“This approach can be considered as a simplified implementation of the Kummerer et al. method for saliency map prediction⁵⁰.”

Q3: I'm unsure what is really learned by the CNN analysis, beyond that the maps are similar (which has already been shown in Figure 1 and 2).

R3: It is true that one the first insights emerging from the CNN analysis is that eye- and digit-tracking maps are similar. But this analysis also shows that both methods can quantitatively measure the levels of influence of the different high-level features on the building of our

visual exploration and that these levels are highly correlated between the two acquisition methods. A standard analysis on the level of similarity of the attention/priority maps cannot show this kind of results. It can also be biased in many way using different post-processing techniques and by the content/complexity of the image database used for the evaluation. The CNN implementation is used here as an image processing filter to identify the different high-level visual features in the image space and to quantify how each feature has been considered on the measured explorations.

Q4: I disagree with specifics of interpreting deep features. The paper adopts the view that the high layers of a deep neural network are selective for single high-level image features (e.g. Figure 3 caption "Each channel can be seen as a saliency map sensitive to a single feature in the picture"). However, this is clearly not the case for CNNs. High layers combine successive nonlinearities rendering easy interpretation intractable. Picking maximally-activating image patches does not necessarily mean a unit's activation is "caused" by whatever is in the patch – hence the large literature on interpretability metrics for CNN features (see e.g. <https://distill.pub/2017/feature-visualization/>, Kim et al (2017)).

R4: True, even if, in the context of the images in our database and for the most responsive channels, the selectivity of the features is quite high. The sentence “*each channel can be seen as a saliency map sensitive to a single feature in the picture*” is slightly overstated. We thank the Reviewer for pointing this out and we now write “*a single feature class*” to remove this ambiguity.

Although we would agree with the Reviewer’s statement that “picking maximally-activating image patches does not necessarily mean a unit’s activation is “caused” by whatever is in the patch”, most of the time it does. Unlike a multilayer perceptron (MLP), the notion of receptive-field exists for CNN and a certain amount of spatial information is preserved throughout the different layers. This phenomenon can be compared to the notion of retinotopy in early cortical visual areas (e.g. Van Essen, 1985). This is the property of CNN that opens the notion of ‘attribution’ (as defined in the Kim et al. 2017 work cited by the Reviewer) and even if the attribution can be more complex than just interpolating a Relu channel (Ancona et al., 2017), it is possible to use this property of CNN to produce very effective saliency models, to make regression tasks, object localization, etc. (e.g. Kümmerer et al., 2014; Vig et al., 2014; Yuan et al., 2014). Thus, making patches of a unit activation is useful because it means that this group of pixels, in the particular context of the input image, contains important information for this channel. But it’s also true to say that the same group of pixels in another context will not activate this channel at all. In other words – a tree in a desert is salient, but the same tree in a forest will not be “considered” a tree, but a part of this forest, or just a part of the environment where another element has to be detected and considered (e.g. an animal). In this case, the tree will activate another channel (with a negative weight in our study) that is also extremely important to quantify and understand the optimization of the observed visual explorations. Again, a parallel with neurophysiology can be done where a percept seemingly emerges from ‘nowhere’ just because the context is prone to create it, and is signaled by visual neurons responding to illusory contours in stimuli such as the Kanizsa Triangle. But most of the time the elements in the foveal area are the ones that contributes most to perception.

Q5: Because of (1) and (2), I'm unsure what novel scientific insights the paper provides. The digit tracking technique seems like a useful proxy that could yield interesting new

possibilities for data collection. This paper shows some proof of concepts of this. But if I try to summarise what I now know after reading this paper, I am left with "digit tracking on a touchscreen could be used to measure spatial priority maps as a complimentary technique to eye tracking and mouse tracking (SALICON)." The lack of new scientific insights make me feel this paper may have greater impact at a journal focussed on methods.

R5: We agree with the Reviewer that digit-tracking is mainly a measure of spatial exploration and attention but we believe that it deserves more than a conditional ("could be used") given the evidence presented. We also hope that the new data in this revised manuscript demonstrates that our digit-tracking implementation is different from SALICON, especially with regard to its lower susceptibility to noise in individual data. In addition to the testing of normal human adult participants with the digit-tracking approach, our study further differs from the Jiang et al. study in that it (i) compares directly the performance of digit-tracking (and now mouse-tracking) to the eye-tracking gold standard, (ii) does the same in a clinical population of autistic individuals (iii) provides evidence for the usefulness of a mobile and calibration-free approach in early detection and differential diagnosis of autism and (iv) highlights its potential in other domains as well (neuropsychological assessment of cerebral lesions, animal testing).

Specific comments / questions

Q6: * N = 11 seems like a very low number of subjects to be able to reliably estimate inter-subject correlation.

R6: Inter-subject correlation is a robust marker of the variability of exploration density maps that can be estimated with a number well below 11 subjects. In order to convince the Reviewer we have replicated the analyses presented in the manuscript for 11 subjects with populations reduced from 2 to 10 subjects. The results obtained are illustrated in the two figures below.

The first figure represents one example of ISC calculation with a random subsample of 2 to 11 subjects on eye-tracking data (upper panel), on digit-tracking data (middle panel) and on the differences between the two techniques (lower panel). The distribution of ISC estimates across our picture database are particularly stable, even with a reduced number of subjects – and no substantial differences between the two recording methods can be detected.

The second figure shows ISC differences between Eye-tracking and Digit-tracking according all possible combinations of 2 groups (Eye-tracking and Digit-tracking) of 2 to 11 subjects. The boxplot in the upper panel represents the distributions of the mean ISC differences between Eye and Digit tracking measurements for all possible combinations of 2 to 11 subjects. The middle panel shows the probability that a combination of subjects reveals a significant difference in the ISC between the two recording methods and the lower panel shows the number of possible combinations. We can observe that the higher the number of subjects used to evaluate the ISC, the lower the probability to find a difference between eye-tracking and digit-tracking. These findings indicate that the probability of having a type II error when using two populations of 11 subjects is very low. Thus, if a possible difference exists and have not been identified by our analysis, the effect-size is most likely negligible.

We have also estimated to have failed to measure a significant difference in the ISC due to our sample size (Type II errors).

Q7: * The Convergence analysis does not show how many subjects are needed to obtain "stable attention maps" as written, but rather is another measure of inter-subject consistency. If two subjects performed exactly the same, this would jump up to 100% with only one subject. A better analysis to show how "stable" the attention maps are is to not normalise to the total variance, but rather show how the slope of improvement of a performance metric (e.g. correlation or AUC) changes as more subjects are added. If the metric isn't changing much then that would be evidence that the maps are stable; if the AUC is still increasing after adding the 11th subject, this is evidence that more subjects are required to obtain stable maps. Something like this was presented in Judd et al (2012; Table 1). My worry with the results presented is that readers may erroneously think that they can make reasonable conclusions after measuring data from only 5 subjects.

R7: The convergence analysis used in our study corresponds to what an experimenter interested in monitoring human behavior generally needs to predict when choosing a population size for her/his experiment in order to observe reliable and stable behavior that can be generalized to an entire population. If two subjects randomly sampled from a population have identical performances, the measured variable will indeed jump to 100% with one added subject and the mean behavior of the whole population can be estimated with only one subject.

An image with a limited number of strongly salient elements will trigger stereotyped exploration behaviors in the general population, which is a great advantage of the method, particularly for diagnostic purposes. The reader would not make a mistake by concluding that a reasonable estimation can be made about the localization of the most salient areas in a scene from a limited number of subjects. Of course, if the expected effect explains a very small

amount of the total variance of the exploration densities, a larger population will be needed to highlight this effect.

Nevertheless, we concede that the normalization step could mask some differences between methods. Thus, as recommended, we have computed the metric proposed by Judd et al. (the quality of prediction of n subjects for n other subjects), limited to how 5 subjects can predict the exploration of 5 other subjects due to our population size and our experimental design (a maximum of 11 subjects have explored the same pictures with the same recording mode). The results are shown in a new supplementary figure S2:

Supplementary Figure S2: Convergence analysis according to the method of Judd et al. 2012. Each curve represents, for each picture, how much an attention map derived from N subjects correlates with another attention map derived from N other subjects. The figures on the left represent this analysis for digit-tracking compared with the standard eye-tracking method. No differences between the two recording strategies were found using this analysis (Uncorrected sign test p -values all >0.05).

Q8: * The paper continuously uses the word "hierarchical" to describe the results of the CNN analysis. The CNN is hierarchical but the analysis presented uses only one layer; the authors rather seem to use this to refer to their rank-ordered linear weights for the different feature maps. This is just a linear combination, which is in no sense "hierarchical" (e.g. the second weight is not determined from the first weight).

R8: Yes, we use “hierarchical” to characterize the relative attentional priority or weight assigned to visual features, not in reference to neural network organization. This is a commonly used terminology, even in the artificial vision literature (Vig et al. 2014).

Q9: * p. 7 "it is possible to detect among the essential features that combine to create almost all possible images (and our perception of these images)". This is overstated. How does AlexNet (1) create almost all possible images and (2) explain how the "essential features" combine to create "our perception of these images"?

R9: Not “almost all” but more than one million. It is an overstatement from a computer vision standpoint, but for a biological visual system this sample is probably sufficiently large sample to learn about the majority of objects encountered in the real world. The sentence has been modified:

“it is possible to detect among the essential features that combine to create a large range of images (and our perception of these images), the hierarchical ordering of features that drive visual attention.”

Q10: * What does it mean if a high-functioning autistic person is "rehabilitated"?

R10: High-functioning does not mean “cured”. It is a diagnostic label for individuals with normal or above-normal IQ at the time the diagnosis of autism is made. The patients that we tested were recruited from a center where they had been receiving help for many years. We have formulated this more explicitly in the revised manuscript:

“The sensitivity of the eye and digit-tracking approaches for the diagnosis of autistic spectrum disorders (ASD) was assessed using attention maps obtained for 22 High-Functioning autistic patients who had undergone extensive rehabilitation and 22 matched control subjects. We reasoned that if a digit tracking-based procedure could detect individuals who had been trained and may have developed coping strategies to overcome their social interaction difficulties, it would make a strong argument for its potential as a diagnostic tool.”

Q11: * Discussion: The discussion on page 10 and 11 veers into rambling incoherency. For example, the Young-Helmholtz trichromatic theory is cited as an "example in the domain of artificial perception"?

R11: The allusion to the Young-Helmoltz theory was introduced as an example of a biological theory that has inspired engineering developments, and in this particular case color photography. We clarified this in the relevant passage of the discussion and hope this will satisfy the Reviewer:

“Examples of biomimetic approaches in the domain of artificial perception are numerous, from the Young-Helmholtz theory of trichromatic color vision (Helmholtz, 1867; Maxwell, 1857; Young, 1802) that spurred the invention of the color camera to the most recent developments in deep learning with convolutional neural network that are mimicking the layers of the visual cortex (LeCun et al., 2015), via the Perceptron (Rosenblatt, 1958) and the solving of multi-sensors mixing problems through Blind Source Separation (Jutten and Herault, 1991).”

Q12: * p.13: Eyetracking accuracy not discussed. I am worried from the example in Figure 1 that the calibration accuracy was not very good (e.g. Figure 1C, heatmap is not centered on the cat's eye)

R12: Fixation on the inner corner of the eye during face scanning is a very common behavior (e.g. <http://doi.org/10.24867/JGED-2016-1-005>) and does not reflect poor calibration. Poor or loss of calibration during the experiment usually causes a systematic bias in the same direction at the subject level, but these biases are in practice eliminated/greatly reduced in the group estimates. We are confident about calibration quality in our study. The following details about the procedure have been added in the Methods section:

“Recordings were made by experienced specialists familiar with the equipment used and with the running of cognitive experiments in normal and patient populations. The stability of the calibration and the behavior of the subject was constantly checked by the operator during the experiment by, in real time, observing if the fixations on the fixation-cross was correctly realized and observed before each picture presentation. If any problems were detected during the recording, the data were excluded from the analysis in order to not bias the quality and stability of measurements. Using these procedure and experimental setup, the usually observed mean deviation between real and measured focalization point is in the order of one degree of visual angle⁴⁴ (Morgante et al., 2012).”

Q13: * p. 14: A "probability density" sums to 1, but the densities used in the study are normalised to have a max of 1. Correct language throughout paper.

R13: OK. Changes made in the text and in the method section.

Q14: * Fixation densities were estimated with a kernel bandwidth of 30 pixels for both eye data and digit data. How was this selected? How do the results depend on this choice? Given that the screen resolution on the touchscreen was far higher than the eyetracking video monitor, does the 30 px bandwidth correspond to the same visual angle in both cases, or different? If different, does this create a problem for the comparison?

R14: Eye-tracking and Digit-tracking data were transposed in the original image space (picture maximized to fit in a 1280x1024 pixels rectangle) optimized for the eye-tracker. The 30-pixel bandwidth therefore corresponds to the same smoothing parameters for attention maps calculation and approximately the same visual angle in both recording methods ($\approx 1^\circ$ of visual angle). This order of magnitude also corresponds to the level of inaccuracy of pictures explorations recorded in eye-tracking studies (Morgante et al., 2012) and has therefore been retained for this study. In order to make this particular point clearer for the reader, the two ‘offline processing’ sections for respectively the Eye-tracking and the digit-tracking methodological sections have been merged into a single paragraph in the “Recording methods comparisons” section:

“Offline processing: First fixations were identified in the continuous eye-gaze data using a fixation filter (Fixation radius = 50pixels, minimum duration = 50ms). Only fixations in the picture area were considered for analysis. Then, for digit-tracking, the regularly sampled location of the center of the simulated foveal windows was transposed in the original picture space to ensure that all analyses are done in the same space with the same spatial resolution

for the two methods (picture maximized to fit a 1280x1024 pixels rectangle). For each subject and each picture, the probability density of exploration, or attention map, was calculated from the fixations data using kernel density estimation with a Gaussian kernel weighted by fixation duration and a bandwidth of 30 pixels ($\approx 1^\circ$ of visual angle) in order to compensate for the usual imprecisions encountered with these methods (e.g.⁴⁸). Finally, each density was scaled between 0 and 1 by dividing the whole density by the maximum density value and group estimates were calculated by averaging scaled densities recorded at the subject level and scaled using the same unity-based normalization procedure.”

Q15: * p. 15: "The sensitivity to the noise of a recording method can be estimated using inter-subject correlation analysis" -- this analysis would confound the noise of the method (precision of estimating spatial attention location) with the consistency of the subjects. You could have a very precise measurement device but a low inter-subject correlation if people are inconsistent (or vice versa, consistent subjects measured with a poor device will show lower correlation than they should). Re-word.

R15: The inter-subject correlation analysis confounds different sources of noise, including the method, the subjects and the image content. That is why we used a counterbalanced design to deal with the confounding variables and focus the analysis on the precision of the two recording methods. Additional references to the fact that the study was designed with a counterbalanced design have been included in the revised main text (highlighted in red). Note that neurotypical subjects are most of the time particularly consistent in their explorations – with an optimized behavior toward the most salient object in the scene. The principal source of variability for the ISC is across images, and that is why the number of evaluated images is far more important.

Q16: * The optimisation of the linear weighting function for combining AlexNet feature maps contains more parameters than data. Discuss.

R16: We disagree with the Reviewer. There are more data than parameters since the output of each channel of the relu5 layer is not unidimensional but multidimensional (13x13 pixels before the interpolation) – 227x227 after the interpolation. Therefore, the regression model has 227x227x122x256 (interpolated CNN channels size x number of pictures x number of channels in the ReLu5 layer) data points to predict 227x227x122 values (Measured attention maps size x number of pictures) with 256 coefficients. Coefficients are not estimated from “pictures” but from small areas in these images that constitutes the receptive fields of the last convolutional layer of the AlexNet architecture.

It is true that the data points are highly correlated and generally, more than 122 pictures are necessary to obtain robust regression models. Unfortunately, the number of images that a subject can explore is necessarily limited, so optimizing the regression method is not a factor that can be neglected. This point was discussed in the supplementary section and two solutions were proposed and evaluated (wCorr and SVMregress).

Q17: * p. 17: "This network was trained twice, using empirically derived eye-tracking data first, then using digit tracking". This sentence reads as if it's a single network being trained sequentially on both datasets, but I think rather you've trained two independent networks?

R17: Yes, the Reviewer is absolutely correct. We rephrased the sentence as follows:

"Two networks with the same architecture were trained, one using eye-tracking data, and the second using digit tracking data"

Q18:* p. 19: "non-parameteric distribution fitting tool" -- how is it non-parametric if it assumes a Gaussian kernel density?

R18: We kindly refer the Reviewer to this link:

https://en.wikipedia.org/wiki/Kernel_density_estimation : "In statistics, kernel density estimation (KDE) is a non-parametric way to estimate the probability density function of a random variable."

Q19: * Figure 2C: in the inter-subject correlations, it seems that each subject contributes more than one data point -- which would make these distributions no longer independent. Discuss in relation to the statistical tests of correlation coefficients / rank-tests, which assume independence.

R19: Individual data points in this analysis represent the mean ISC for each image. Thanks to the counterbalanced design, the images were explored by different subjects with each tracking method, hence the distributions of ISC scores for the two methods can be considered independent from one another.

Q20: * Figure 5: caption is confusing, and could be misunderstood to mean that the plots in B show data from only four subjects.

R20: We fixed this by specifying the Ns in the relevant sentence of Figure 5 legend:

"For each subject, a single score was calculated to quantify the neurotypicity of attention maps obtained in patients (N=22) and control subjects (N=22)."

Q21: * Figure 5C: if I understand correctly the classification is based on 22 ASD (20 for Eyes condition) and 22 Ctrl subjects. If this is the case, I would expect the ROC curves to be less smooth -- they should jump up / right as threshold changes.

R21: The smoothness is explained by the fact that we first fitted a non-parametric distribution then computed the ROC, as already described in the Results section:

"In order to compare the discriminatory power of the eye-tracking approach with that of digit-tracking, we first used a non-parametric distribution fitting tool to estimate the distributions of the score in both populations. Then, we calculated the Receiver Operating Curves (ROC) derived from the estimated distributions to finally obtain evaluations of the classification performances using the AUC (Area Under the Curve)."

Q22: * Figure S1C: The analysis of the SVMregress coefficients seems misguided. The Lasso penalty forces all the weights to be really small (compare scales of top and bottom

scatterplots). So all these weights are basically just at a point in the middle of the top plot -- ie. they've all been set to near-zero. In this case, I'm not even sure why this model works at all - it seems the regularisation penalty is too strong.

R22: SVMregress and wCorr are two methods that are conceptually very different and therefore the scales of the two scatter plots in the figure in the Supplementary section are not comparable. We use a cross-validated strategy to tune the LASSO regularization penalty and assure the best result for the SVMregress method. The coefficients, learned with an SVM regression on the subset A and B, with and without an optimized LASSO regularization, are plotted in the following scatter plots that are here, comparable:

SVM regression coefficient learned without (left) and with (right) a cross validated LASSO regularization.

Without the cross-validated LASSO regularization, the correlation between coefficients learned in the datasets A and B is poor. After the cross-validated regularizations, no coefficient was equal to zero, but a substantial shrinkage of the coefficients is realized. The correlation between coefficients learned on the two image datasets are far better but still much lower than the correlation obtained with the wCorr method.

More information has been added in the method section in order to highlight the procedure selected for optimizing the regularization parameters (Saliency map prediction via CNN regression section, page 18).

“A learning strategy optimized for high-dimensional data was selected considering the important number of parameters estimated and the correlation in the data after the interpolation step (256 regression parameters, learned using fitrlinear – Matlab r2017a (Mathwork Inc.) – with a 10-fold cross-validated lasso regularization).”

Q23: ## Open data and materials

The primary data, materials and code upon which this manuscript is based do not seem to be publicly available. I strongly encourage the authors to either

1. Make the data and materials publicly available in a reliable third-party repository (for example, the Open Science Framework -- see <http://osf.io/> or Zenodo (<http://zenodo.org/>)).

OR

2. State in the manuscript method section their reason(s) for not doing so.

If the authors choose to do (1), there are many more details and guidelines on how to do this at <https://opennessinitiative.org/the-initiative/>. Not only is this a great move in support of open science and transparency, but it may even mean the paper gets cited more (see for example Piwowar & Vision, 2013 or McKiernan et al 2016). It will certainly facilitate any future secondary analyses.

If the authors choose (2), note that there are many legitimate reasons why (1) cannot be done. For example, the authors may not be able to effectively anonymise data, or they do not own the copyright to the data or materials. In these cases, analysis and experiment code can almost always be released, but the degree to which the authors comply is entirely up to them. I require only that they state in the manuscript why they have not.

Should the authors be unwilling to do (1) or (2), then I cannot recommend the paper for publication because it does not meet the minimum quality requirements for a scientific manuscript.

R23: We are sensitive to the Reviewer's arguments about open science. In the original submission of the paper we disclosed a "conflict of interest" as our Institution has filed for intellectual property rights on the digit tracking method in the event it would be used for commercial purposes. As the patent has now been published, there are no obstacle to making the data contained in the manuscript publicly available as well as the Matlab script necessary to run the digit tracking experiment described, in conjunction with the Matlab Psychtoolbox-3. The following statement has been inserted following the Methods section in the revised manuscript:

"Data on healthy subjects and Matlab scripts are available on Open Science Framework <http://osf.io/>. Patients' data are available on request."

Reviewer #3 (*Remarks to the Author*):

The authors introduce a new tactile interface for visual perception analysis. This new method is clever and seems to have potential value in a restricted number of domains.

Q1: My most important concern reflects the lack of open science of this project. The whole manuscript reads like a brochure for a new software tool and, given the patent that was filed, it seems like a matter of time before this technology is transformed into a commercial product. Especially because this manuscript is pre-dominantly a methodological contribution (new theoretical is lacking), I would only want to advise Nature Communications to publish this type of paper if the software is open source and freely available. To me, open source is key in bringing the scientific endeavor forward.

R1: We are sorry that our presentation of the potential applications of digit-tracking made the Reviewer feel like reading a sales pitch. We felt that a paper focusing on a new method should demonstrate what can be accomplished with it. Digit-tracking may eventually find its way in commercial applications since there is still obviously a market for new eye tracking

technologies. But the present manuscript focuses on research and we would argue that it makes a scientific contribution even though it presents more a methodological than a conceptual advance.

The Reviewer rightly argues in favor of open science and as stated above in response to Reviewer 2, we will post the data and make available to researchers upon request a set of functions allowing to run a digittracking experiment in conjunction with the Matlab Psychtoolbox-3.

Q2: Besides this main point, I think the authors should be more clear in stating for which cognitive domains this technique is interesting. It's definitely not a replacement for eye tracking as a whole (which is how the manuscript is currently framed), because eye tracking is widely used in all types of domains. Here, scene viewing is convincingly demonstrated, but I don't see this technique play a role in measuring oculomotor competition, such as in the antisaccade or oculomotor capture task, or visual constancy, such as double-step and landmark tasks.

R2: We agree with the Reviewer that our technique is an alternative to eye-tracking in the restricted case of scene exploration analysis. Its limitations were already acknowledged in the first paragraph of the Discussion section in the original manuscript and now, in response to the similar point made by Reviewer 1, we added a few caveats:

“As an indirect measuring device, digit-tracking is obviously excluded from some applications of optical eye-tracking, like calculating the dynamics (e.g. speed, acceleration) of reflexive and voluntary eye movements (e.g. optokinetic and vestibulo-ocular reflexes, smooth pursuit, pro- and antisaccade, oculomotor capture, etc.), blink rate or pupil size.”

Q3: The authors report a reliable correlation between eye-tracking and digit-tracking, but only present the image for 4 seconds (in the eye-tracking version). Because of experimental constraints (e.g. central fixation point), exploration starts at the centre of the screen. It should therefore be checked whether this correlation also holds for the final two seconds and is not driven by the initial movements.

R3: The Reviewer makes an important point, as this central bias is common in eye-tracking experiments. In our study, subjects looked at a fixation cross at the center of the screen and pressed a key on a keyboard to initiate picture presentation. In order to reduce the influence of this central bias in our data, only fixations that followed the first saccadic movement were considered. Nevertheless, at the moment of image appearance, subjects had their foveal vision in the center of the screen and this initial condition could have biased their exploration behavior toward the higher-resolution perifoveal locations at the expense of lower-resolution peripheral locations. In the Digit-Tracking protocol, subjects pressed a virtual button at the center of the screen and picture presentation started upon button release. Here again, the subjects' fovea is initially placed at the center of the screen. However, because of the blur, image resolution is homogenous across the entire retina and the center is no more salient than the periphery. The Reviewer's remark prompted us to reprocess our data, adding a third dimension of exploration length (expressed as % total exploration time for eye-tracking and as % of total exploration distance for digit-tracking) in order to determine how the construction of attention maps evolves and consequently how exploration length could influence measures of inter-subject correlation (ISC) and of consistency between Eyes and

Digit-derived maps. To do so, eye-tracking and digit-tracking measurements were transposed in a 3D space of $100 \times X$ pixels in space, where $X = \max(\text{Height}, \text{Length}) / \min(\text{Height}, \text{Length}) * 100$ x 100 data points for exploration length (i.e. we decreased the exploration map resolution, while respecting width to height ratio of each image, in order to limit the computational load). These attention maps were processed using a 3d smoothing Gaussian kernel with a $\sigma=3$ in space and a $\sigma=1$ in time. Then the cumulative sum of the maps was computed along the time dimension and normalized for each subject and at each time point by dividing the estimated densities by the maximum value. This procedure allowed to estimate the build-up of exploration maps along the exploration length.

The results of this analysis are included in the revised manuscript in a new supplementary figure S1. The top plot (A) shows significantly higher inter-subject correlations for eye-tracking than digit-tracking in the early stages of exploration (1 to 17% of exploration length). This is consistent with the Reviewer's prediction that initial ISC may be inflated by a central location bias which, however, is not observed in the digit-tracking data. However, this bias vanishes as data accumulate (exploration length $> 34\%$, $\approx 1.5s$ of eye-tracking data) and has a negligible impact on the final ISC estimates. The bottom plot (B) shows that the correlation between eye- and digit-tracking attention maps are lowest below 20% of exploration length, most likely due to fact that the data obtained by the two methods are not biased in the same way at the early stages of exploration. However, this initial difference also has little impact on the final estimate of correlation between eye- and digit-tracking.

Figure S1. Effects of exploration length on saliency map consistency. In order to determine how attention maps evolve over time and examine the impact on exploration statistics of a potential initial central bias due to the presence of a fixation cross (eye-tracking) or start button at the screen center, we took into account a third dimension of exploration length (expressed as % total exploration time for eye-tracking and as % of total exploration distance for digit-tracking). In order to limit the computational load, eye-tracking and digit-tracking measurements were transposed in a 3D space of $100 \times X$ pixels in space, where $X = \max(\text{Height}, \text{Length}) / \min(\text{Height}, \text{Length}) * 100$ x 100 data points for exploration length (i.e. we decreased the exploration

map resolution, while respecting width to height ratio of each image). These attention maps were processed using a 3d smoothing Gaussian kernel with a $\sigma=3$ in space and a $\sigma=1$ in time. The cumulative sum of the maps was then computed along the time dimension and normalized for each subject and at each time point by dividing the estimated densities by the maximum value. This procedure allowed to estimate the build-up of exploration maps along the exploration length. **A)** Cumulative inter-subject correlations (ordinate) were computed for each picture, with digit-tracking or eye-tracking, from 1% to 100% of exploration length (abscissa). Inter-subject correlations significantly higher for eye-tracking than digit-tracking in the early stages of exploration (1 to 17% of exploration length). This suggests that initial ISCs are inflated by a central location bias which, however, is not observed in the digit-tracking data. However, this bias vanishes as data accumulate (exploration length > 34%, $\approx 1.5s$ of eye-tracking data) and has a negligible impact on the final ISC estimates. **B)** Correlation between Eyes and Digit derived attention maps, from 1% to 100% of exploration. The correlations between eye- and digit-tracking attention maps are lowest below 20% of exploration length, most likely due to fact that the data obtained by the two methods are not biased in the same way at the early stages of exploration. However, this initial difference also has little impact on the final estimate of correlation between eye- and digit-tracking. Thin lines in both plots show cumulative values of individual images and the thick line represents the group average.

Q4: They should also examine whether measures of search efficiency correlate between both tracking methods.

R4: This is an interesting suggestion that will need to await future work. Adequate estimation of search efficiency requires special-purpose visual search arrays and task instructions (i.e. reporting the presence of a target among distractors). Our experiment investigated spontaneous scene exploration and are not well-suited to address this issue.

Q5: The case descriptions provide little information. If the authors really want to make a case for e.g. visuospatial neglect, they should test a full patient group.

R5: We presented these single cases for illustrative purposes because they are representative of a consistent group tendency. This was shown for the stereotypic autistic behavior in the original version. In order to strengthen the case for hemispatial neglect detection with digit-tracking, we now present results for a new group of right brain-damaged patients ($n=4$) in a new supplementary figure S5. Although this is a small sample, the visual exploration asymmetry is highly significant (consider the z-scores reported in Fig S5) and representative of a classical neuropsychological syndrome that has been extensively investigated.

Figure S5. Representative examples of the mean density of exploration obtained from patients with right ischemic or hemorrhagic strokes. **A) Left:** Mean spatial distribution of digit-tracking explorations for a population of 22 control subjects. **Right:** Mean spatial distribution of digit-tracking explorations for a population of 4 patients, with right ischemic or hemorrhagic lesions. Note the tendency to neglect the left side of the pictures. **B)** Individual maps for each of these 4 patients. **P1:** Right ischemic stroke – Asymetry right = 55.8%, $Z=2.39$. **P2:** Right ischemic stroke (Middle cerebral artery) – Asymetry right = 61.4%, $Z=4.07$. **P3:** Right ischemic stroke (Middle cerebral artery – total) – Asymetry right = 71.8%, $Z=7.18$. **P4:** Right hemorrhagic stroke – Asymetry right = 77.9%, $Z=9.03$.

References:

- Ancona, M., Öztireli, C., Ceolini, E., Gross, M., n.d. A unified view of gradient-based attribution methods for Deep Neural Networks 11. 31st Conference on Neural Information Processing Systems (NIPS 2017)
- Esteve-Gibert, N., Prieto, P., 2014. Infants temporally coordinate gesture-speech combinations before they produce their first words. *Speech Commun.* 57, 301–316. <https://doi.org/10.1016/j.specom.2013.06.006>
- Helmholtz, H. von, 1867. *Handbuch der physiologischen Optik*. Leipzig : Leopold Voss.
- Jiang, M., Huang, S., Duan, J., Zhao, Q., 2015. SALICON: Saliency in Context, in: 2015 IEEE Conference on Computer Vision and Pattern Recognition (CVPR). Presented at the 2015 IEEE Conference on Computer Vision and Pattern Recognition (CVPR), pp. 1072–1080. <https://doi.org/10.1109/CVPR.2015.7298710>
- Jutten, C., Herault, J., 1991. Blind separation of sources, part I: An adaptive algorithm based on neuromimetic architecture. *Signal Process.* 24, 1–10. [https://doi.org/10.1016/0165-1684\(91\)90079-X](https://doi.org/10.1016/0165-1684(91)90079-X)
- Kümmerer, M., Theis, L., Bethge, M., 2014. Deep gaze i: Boosting saliency prediction with feature maps trained on imagenet. *ArXiv Prepr. ArXiv14111045*.
- LeCun, Y., Bengio, Y., Hinton, G., 2015. Deep learning. *Nature* 521, 436–444. <https://doi.org/10.1038/nature14539>
- Maxwell, J.C., 1857. XVIII.—Experiments on Colour, as perceived by the Eye, with Remarks on Colour-Blindness. *Earth Environ. Sci. Trans. R. Soc. Edinb.* 21, 275–298. <https://doi.org/10.1017/S0080456800032117>
- Morgante, J.D., Zolfaghari, R., Johnson, S.P., 2012. A Critical Test of Temporal and Spatial Accuracy of the Tobii T60XL Eye Tracker. *Infancy* 17, 9–32. <https://doi.org/10.1111/j.1532-7078.2011.00089.x>
- Rosenblatt, F., 1958. The perceptron: A probabilistic model for information storage and organization in the brain. *Psychol. Rev.* 65, 386–408. <https://doi.org/10.1037/h0042519>
- Van Essen, D. C. Functional organization of primate visual cortex. In *Cerebral Cortex* Vol. III, A. Peters and E. G. Jones (Editors), pp. 259–329. Plenum, New York, 1985.
- Vig, E., Dorr, M., Cox, D., 2014. Large-Scale Optimization of Hierarchical Features for Saliency Prediction in Natural Images, in: 2014 IEEE Conference on Computer Vision and Pattern Recognition. Presented at the 2014 IEEE Conference on Computer Vision and Pattern Recognition (CVPR), IEEE, Columbus, OH, USA, pp. 2798–2805. <https://doi.org/10.1109/CVPR.2014.358>
- Young, T., 1802. II. The Bakerian Lecture. On the theory of light and colours. *Philos. Trans. R. Soc. Lond.* 92, 12–48. <https://doi.org/10.1098/rstl.1802.0004>
- Yuan, J., Ni, B., Kassim, A.A., 2014. Half-CNN: a general framework for whole-image regression. *ArXiv Prepr. ArXiv14126885*.

REVIEWERS' COMMENTS:

Reviewer #1 (Remarks to the Author):

The authors have satisfactorily addressed my previous comments.

Reviewer #3 (Remarks to the Author):

Although I like the newly developed method and I'm happy to read that the authors are making (part of) their files open access, I still believe there is too little theoretical advancement to warrant publication in Nature Communications. There is no knowledge in this paper that has theoretical value. I think a journal focused on methods is a better outlet for this paper.